# Marginal Causal Flows for Validation and Inference

**Daniel de Vassimon Manela**[*]
University of Oxford
manela@stats.ox.ac.uk

**Laura Battaglia**[*]
University of Oxford
battaglia@stats.ox.ac.uk

**Robin J. Evans**
University of Oxford
evans@stats.ox.ac.uk

## Abstract

Investigating the marginal causal effect of an intervention on an outcome from complex data remains challenging due to the inflexibility of employed models and the lack of complexity in causal benchmark datasets, which often fail to reproduce intricate real-world data patterns. In this paper we introduce Frugal Flows, a novel likelihood-based machine learning model that uses normalising flows to flexibly learn the data-generating process, while also directly inferring the marginal causal quantities from observational data. We propose that these models are exceptionally well suited for generating synthetic data to validate causal methods. They can create synthetic datasets that closely resemble the empirical dataset, while automatically and exactly satisfying a user-defined average treatment effect. To our knowledge, Frugal Flows are the first generative model to both learn flexible data representations and also *exactly* parameterise quantities such as the average treatment effect and the degree of unobserved confounding. We demonstrate the above with experiments on both simulated and real-world datasets.

## 1 Introduction

Simulating realistic datasets such that the marginal causal effect is constrained to take a specific form is a significant challenge in causal inference. Many methods for inferring these effects exist, but simulating from them is a significant challenge (Young et al., 2008; Havercroft and Didelez, 2012; Keogh et al., 2021). In particular, it is difficult to simulate complex benchmarks from generative models in such a way that a custom marginal effect exactly holds.

The *frugal parameterisation* (Evans and Didelez, 2024) provides a solution to this problem by constructing a joint distribution that explicitly parameterises the marginal causal effect and builds the rest of the model around it. Frugal models typically represent the dependency between an outcome and pretreatment covariates using copulae. Standard multivariate copulae are parametric, leading to potential model misspecification.

In this paper we show how one can construct frugally parameterised marginal causal models using normalising flows (NFs, Rezende and Mohamed, 2015; Dinh et al., 2016) to target the causal margin of the distribution (a conditional univariate marginal density of an outcome conditioned on a treatment). We name the resulting model a *Frugal Flow* (FF). To the best of our knowledge, FFs offer the first likelihood-based framework for learning a marginal causal effect while modelling the outcome and propensity nuisance parameters using flexible generative models.

FFs are exceptionally well suited for generating benchmark datasets for causal method validation. Since FFs enable direct parameterisation of the causal margin, they provide a framework for generating causal benchmark datasets which resemble real-world datasets, but which also allow users to encode causal properties in order to validate novel inference models. FFs can be used to generate benchmarks with customisable degrees of unobserved confounding. This can aid in the validation

---

[*]Equal Contribution

of model robustness under conditions where the assumption of conditional ignorability does not hold. Here, conditional ignorability (or conditional exchangeability) means that marginal distribution of the potential outcomes is independent of the value of treatment, conditional on the observed covariates (Pearl, 2009).

FFs offer marked improvements over current benchmarking generation methods, which use soft constraint optimisation to enforce the desired causal restrictions (Kendall, 1975; Parikh et al., 2022). As a result, *post hoc* checks are required to see whether these conditions are present in the synthetic data. FFs do not require this second step, as relevant conditions are explicitly encoded in the underlying likelihood. Finally, FFs allow for outcomes to be sampled from marginal logistic and probit models, making them the first generative benchmarking model to facilitate the simulation of binary outcomes with a choice of user specified risk differences, risk ratios, or odds ratios.

## 2  Background

In this paper we consider a static treatment model with an outcome $Y \in \mathcal{Y} \subseteq \mathbb{R}$ and $T$ a binary treatment in $\mathcal{T} = \{0, 1\}$. Let the set of measured pretreatment covariates be $\boldsymbol{Z} \in \mathcal{Z} \subseteq \mathbb{R}^D$. Additionally, we will use the notation of Pearl (2009) where intervened distributions are indicated by the presence of a "$\mathrm{do}(\cdot)$" operator, with its absence indicating that the distribution is from the observational regime.

### 2.1  Marginal Causal Models

Causal inference methods are generally developed to estimate the average effect of a treatment $(T)$ on an outcome $(Y)$ for a population defined by a set of pretreatment covariates $(\boldsymbol{Z})$ (Hernán and Robins, 2020). Let the variables be distributed according to $(\boldsymbol{Z}, T, Y) \sim P_{\boldsymbol{Z}TY}$ with density $p_{\boldsymbol{Z}TY}$. We make the standard assumptions of a stable unit treatment value (commonly referred to as SUTVA), positivity, and conditional ignorability (equivalent to conditional exchangability) outlined in Pearl (2009). Additionally, the covariate set $\boldsymbol{Z}$ must only include pretreatment covariates. The conditional distribution of $Y$ and $\boldsymbol{Z}$ after an intervention on $T$ is equal to

$$p_{\boldsymbol{Z}Y|\mathrm{do}(T)}(\boldsymbol{z},\ y \mid t) = p_{\boldsymbol{Z}}(\boldsymbol{z}) \cdot p_{Y|\boldsymbol{Z},\mathrm{do}(T)}(y \mid \boldsymbol{z},\ t).$$

Causal practitioners are often interested in the marginal effect of $T$ on $Y$ on the intervened system, sometimes referred to as the marginal outcome distribution (MOD), $p_{Y|\mathrm{do}(T)}$:

$$p_{Y|\mathrm{do}(T)}(y \mid t) = \int_{\mathcal{Z}} d\boldsymbol{z}\, p_{Y|\boldsymbol{Z},\mathrm{do}(T)}(y \mid \boldsymbol{z}, t)\, p_{\boldsymbol{Z}}(\boldsymbol{z}). \tag{1}$$

The difference between the means of $Y$ under this margin between different values of $T$ is called the average treatment effect (ATE), $\tau$ where, $\tau = \mathbb{E}[Y \mid \mathrm{do}(T = 1)] - \mathbb{E}[Y \mid \mathrm{do}(T = 0)]$. Models which target this marginal quantity are known as *marginal structural models* (MSMs, Robins, 1998) and are frequently used in epidemiological and medical domains to account for time-varying confounding. In particular, they are effective at quantifying the effect of an intervention over a population, where the specific relationships between the outcome and (possibly high dimensional) pretreatment covariates are not relevant, and are modelled as nuisance parameters. The semiparametric question of estimating finite dimensional quantities in the presence of high dimensional nuisance parameters has a long history (Robins et al., 1995; Robins and Rotnitzky, 1995), but has undergone a renaissance since the development of methods such as targeted maximum likelihood estimation (van der Laan and Rose, 2011) and double machine learning (Chernozhukov et al., 2018), which allow for general machine learning algorithms to flexibly describe the nuisance models and still have valid inference on a low-dimensional treatment effect.

### 2.2  Frugal Parameterisations

Frugally parameterised distributions consist of three distinct components: the distribution of the 'past,' $\theta_{ZT}$; the intervened causal quantity of interest, $\theta_{Y|\mathrm{do}(T)}$; and an intervened dependency measure between $Y$ and $\boldsymbol{Z}$ conditional on $T$, $\phi_{\boldsymbol{Z}Y|\mathrm{do}(T)}$. The key idea is to explicitly parameterise the marginal causal effect, and build the rest of the model around it. In this paper we encode all the dependence among covariates in the copula, so 'the past' is really just the propensity for treatment

(also called the propensity score) and the product of the univariate margins (Evans and Didelez, 2024). Figure 1 provides an illustrative summary of our framework, and outline which models are used to parameterise each component of a frugal model.

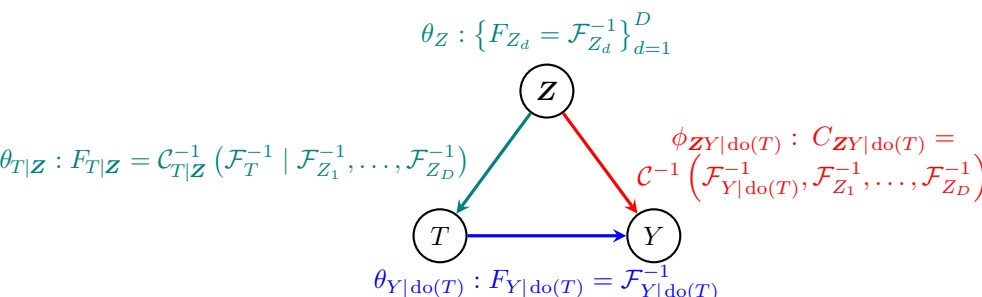

Figure 1: A visual abstract outlining the different components of a frugal model, and how each specific component is parameterised. Univariate CDFs are denoted by $F$, and copula distribution functions are denoted by $C$. The marginal causal effect, $\theta_{Y|\mathrm{do}(T)}$, is modelled with a univariate normalising flow, which we denote by $\mathcal{F}$ (see Section 2.4). The intervened dependency measure, $\phi_{\boldsymbol{Z}Y|\mathrm{do}(T)}$, is modelled with a copula flow which we denote by $\mathcal{C}$ (see Section 2.5). The past, $\theta_{ZT}$, is modelled by the combination of univariate normalising flows (for the univariate pretreatment covariate distributions) and a copula flow (for the propensity of treatment).

**Variation Independence**   Any smooth and regular 'dependency measure' can be chosen for parameterising $\phi_{\boldsymbol{Z}Y|\mathrm{do}(T)}$; this is defined as a quantity which, when combined with the marginal distributions, smoothly parameterises the joint distribution. It is desirable that the three parameter sets $(\theta_{\boldsymbol{Z}T}, \theta_{Y|\mathrm{do}(T)}, \phi_{\boldsymbol{Z}Y|\mathrm{do}(T)})$ are *variation independent* (Barndorff-Nielsen, 2014) of each other; such parameterisations have the benefit of allowing the measure $\phi_{\boldsymbol{Z}Y|\mathrm{do}(T)}$ to be freely specified without restricting the rest of the model. Copulae are an example of such a dependency measure, and are a natural choice for frugally modelling dependencies in continuous and mixed datasets. For further detail we refer the reader to Appendix A.

## 2.3   Copulae

A multivariate copula, denoted by $C : [0,1]^d \to [0,1]$ is a multivariate cumulative distribution function (CDF) defined over a set of $d$ uniform margins, with an associated density $c(\cdot)$ if it is continuous with respect to its arguments (Sklar, 1959; Joe, 2014). Copulae are often used to parameterise the dependency structure of a joint distribution independent of its univariate margins. Large, complex dependency structures are often modelled by pair-copula constructions (PCCs) or vine copulae (Czado and Nagler, 2022; Joe and Kurowicka, 2011). These methods factorise the dependency structure into a set of non-overlapping bivariate copulae. However, these approaches typically impose the constraints of a finite dimensional parameterisation on the dependency structure in the bivariate copulae used. A more comprehensive introduction to copulae can be found in Appendix B.

**Copulae in Machine Learning**   More complex ML models have been developed to more flexibly learn copula distributions. Several alternatives have been proposed, some targeting specific copula classes (Ling et al., 2020; Wilson and Ghahramani, 2010), and others constraining a neural network-based architecture to estimate valid copulae, though often with limited scalability (Zeng and Wang, 2022; Chilinski and Silva, 2020) or using variational approximations (Letizia and Tonello, 2022). However, the most active research area in this field makes use of normalising flows, leveraging their likelihood-based, composable and invertible nature to chain transformations of marginal quantities to the fitting of the copula density.

**Paper Motivation**   A key motivation for this paper is the search for a flexible parameterisation of the copula

$$\phi_{\boldsymbol{Z}Y|\mathrm{do}(T)}(\boldsymbol{z}, y \mid t) = c(F_{Y|\mathrm{do}(T)}(y \mid t), F_{Z_1}(z_1), \dots, F_{Z_D}(z_D)) \qquad (2)$$

between the probability integral transforms of the univariate pretreatment covariates and a conditional univariate quantity which parameterises the causal margin. Evans and Didelez (2024) show that this can be done using parametric copulae, and also prove that it targets the marginal causal rather than the conditional distribution when $\phi_{\boldsymbol{Z}Y|\mathrm{do}(T)}$ is parameterised by a multivariate copula. Consider the multivariate copula for the distribution of $\boldsymbol{Z}$ and $Y$ conditional on $T$:

$$c(F_{Y|T}, F_{Z_1|T}, \ldots, F_{Z_D|T}).$$

For an intervened distribution, all pretreatment covariates $\boldsymbol{Z}$ are marginally independent of $T$, and so the intervened joint density becomes

$$p_{Y|\boldsymbol{Z},\mathrm{do}(T)} = p_{Y|\mathrm{do}(T)} \cdot c(F_{Y|\mathrm{do}(T)}, F_{Z_1}, \ldots, F_{Z_D}),$$

where $p_{Y|\mathrm{do}(T)}$ is the marginal causal effect of $T$ on $Y$. The final propensity score density $p_{X|\boldsymbol{Z}}$ does not affect the marginal densities in the observational model as there is a parameter cut between $p_{T|\boldsymbol{Z}}$ and $p_{Y|\boldsymbol{Z},\mathrm{do}(T)}$ (Barndorff-Nielsen, 2014). However, $\theta_{Y|\mathrm{do}(T)}$ and $\phi_{\boldsymbol{Z}Y|\mathrm{do}(T)}$ are functions of $p_{Y|\boldsymbol{Z},\mathrm{do}(T)}$ and thus should be estimated jointly. If $p_{Y|\mathrm{do}(T)}$ is estimated separately from the copula, the marginal *conditional* effect will be inferred rather than the marginal *causal* effect.

Generative ML methods allow for estimating more flexible and general copulae, but have struggled so far to learn copulae together with conditional univariate quantities. We resolve this problem and design a NF-based copula inference method that allows for these quantities to be estimated jointly as required by the frugal parametrisation (see Section 2.5). The model is then trained on real-world data and used for generating customised causal benchmarks which closely resemble the original dataset.

## 2.4 Normalising Flows

Normalising flows (NFs) (Tabak and Turner, 2013; Rezende and Mohamed, 2015; Dinh et al., 2016) allow for density estimation via learning a diffeomorphic transformation $\mathcal{F}$ that maps the unknown target distribution $p_{\boldsymbol{X}}(\boldsymbol{x})$, $\boldsymbol{x} \in \mathbb{R}^D$ to a simple and known base distribution $p_{\boldsymbol{U}}(\boldsymbol{u})$, $\boldsymbol{u} \in \mathbb{R}^D$, so that when $\boldsymbol{X} \sim p_{\boldsymbol{X}}$ and $\boldsymbol{U} \sim p_{\boldsymbol{U}}$ then $\boldsymbol{U} = \mathcal{F}^{-1}(\boldsymbol{X})$ .

$\mathcal{F}$ is usually a composition of invertible and differentiable transformations $\mathcal{F}_i$ parametrised by neural networks, and is often trained by maximising the log-likelihood of observed $\{\boldsymbol{x}_i\}_{i=1}^N$. This can be conveniently done in closed form exploiting the change of variable formula

$$p_{\boldsymbol{X}}(\boldsymbol{x}) = p_{\boldsymbol{U}}(\mathcal{F}^{-1}(\boldsymbol{x})) \left| \det\left( \frac{\partial(\mathcal{F}^{-1}(\boldsymbol{x}))}{\partial \boldsymbol{x}} \right) \right|, \tag{3}$$

provided that the chosen model for $\mathcal{F}$ allows for efficient computation of the Jacobian determinant $\det(\partial(\mathcal{F}^{-1}(\boldsymbol{x}))/\partial \boldsymbol{x})$. The implementation of $\mathcal{F}^{-1}$ then allows for density evaluation, whereas $\mathcal{F}$ can be used for sampling from the joint.

As for the choice of $\mathcal{F}$, the literature has explored a number of implementations that retain invertibility while allowing for computational tractability of the determinant. See Papamakarios et al. (2021) for an introduction and overview. Our implementation relies on neural spline flows (NSF, Durkan et al., 2019), a particular type of autoregressive flows that will be further illustrated in Section 2.5.

## 2.5 Copula Flows

Our Frugal Flow approach builds upon the copula-based flow model proposed by Kamthe et al. (2021) for synthetic data generation. The authors start by considering a copula $C(F_{X_1}, \ldots, F_{X_D})$ defined over the marginal probability integral transforms $F_{X_1}, \ldots, F_{X_D}$ of a random vector $\boldsymbol{X} = [X_1, \ldots, X_D]$. Assuming the copula density exists, the joint density of $\boldsymbol{X}$ can be written as

$$p_{\boldsymbol{X}}(x_1, \ldots, x_d) = c_{\boldsymbol{X}}\left(F_{X_1}(x_1), \ldots, F_{X_D}(x_D)\right) \cdot \left[ \prod_{d=1}^{D} p_{X_d}(x_d) \right], \tag{4}$$

where $p_{X_d}$ is the marginal density of $X_d$. This factorisation of the density can be similarly induced by a NF that composes $D$ flows $\mathcal{F}_1, \ldots, \mathcal{F}_D$ for the marginal quantities and a flow $\mathcal{C}_{\boldsymbol{X}}$ for the copula.

For the rest of this paper, we will let $\boldsymbol{U} \sim \mathrm{Uniform}[0, 1]^D$ represent a vector of *independent* uniforms, and let $\boldsymbol{V} \sim C$ represent a vector of *dependent* uniforms as a multivariate copula $C$. The generative

procedure for this NF takes samples $U$ from a base distribution of independent uniforms and first pushes them through the copula flow $\mathcal{C}_{\boldsymbol{X}}$, obtaining correlated uniform samples $\boldsymbol{V} = \mathcal{C}_{\boldsymbol{X}}(\boldsymbol{U})$. Then $\boldsymbol{V}$ is mapped through the marginal flows $\mathcal{F}_{\boldsymbol{X}} = [\mathcal{F}_{X_1}, \ldots, \mathcal{F}_{X_D}]$ to obtain the random vector $\boldsymbol{X} = \mathcal{F}_{\boldsymbol{X}}(\boldsymbol{V})$.

The composed flow $\boldsymbol{X} = \mathcal{F}_{\boldsymbol{X}}(\mathcal{C}_{\boldsymbol{X}}(\boldsymbol{U}))$ is also a valid flow, and via the change of variable formula as in eq. (3) it induces a specific factorisation of the density of $\boldsymbol{X}$ Here, we quote the result from Kamthe et al. (2021):

$$
\begin{aligned}
p_{\boldsymbol{X}} &= p_{\boldsymbol{V}}(\mathcal{F}_{\boldsymbol{X}}^{-1}(\boldsymbol{X})) \left| \det\left( \frac{\partial \mathcal{F}_{\boldsymbol{X}}^{-1}(\boldsymbol{X})}{\partial \boldsymbol{X}} \right) \right| \\
&= \left| \det\left( \frac{\partial \mathcal{C}_{\boldsymbol{X}}^{-1}(\mathcal{F}_{\boldsymbol{X}}^{-1}(\boldsymbol{X}))}{\partial \mathcal{F}_{\boldsymbol{X}}^{-1}(\boldsymbol{X})} \right) \right| \left| \prod_{d=1}^{D} \left( \frac{\partial \mathcal{F}_{X_d}^{-1}(X_d)}{\partial X_d} \right) \right|.
\end{aligned}
\tag{5}
$$

As the univariate mapping from a uniform to a random variable is uniquely defined by the CDF, the flows $\mathcal{F}_{\boldsymbol{X}}^{-1} = [\mathcal{F}_{X_1}^{-1}, \ldots, \mathcal{F}_{X_D}^{-1}]$ target the marginal CDFs $F_{X_1}, \ldots, F_{X_D}$. Note how eq. (5) factorises the density of $\boldsymbol{X}$ into a copula density and a product of marginal densities as in eq. (4).

$\mathcal{C}_{\boldsymbol{X}}$ is estimated with a NSF, a NF of the autoregressive flow class. Autoregressive flows (Papamakarios et al., 2017; Huang et al., 2018) factorise $\mathcal{C}_{\boldsymbol{X}}$ as a recursive sequence of univariate conditional flows:

$$
V_1 := \mathcal{C}_1(U_1) \qquad V_d := \mathcal{C}_{d|1\ldots d-1}(U_d \mid V_1, \ldots, V_{d-1}) \qquad 2 \leq d \leq D.
\tag{6}
$$

In principle, since the input $\boldsymbol{U}$ is a vector of independent uniforms, the conditional flows would approximate the inverse of the Rosenblatt transform (Rosenblatt, 1952) and thus be universal approximators if the flows were sufficiently expressive (Papamakarios et al., 2021). The Rosenblatt transform sequentially maps each component $S_d$ of any random vector $\boldsymbol{S}$ with strictly positive density through its corresponding conditional CDF $F_{S_d|S_1,\ldots,S_{d-1}}$, obtaining a vector $\boldsymbol{U}$ of independent uniforms. It is known to be a diffeomorphism, so its inverse bears the same structure as eq. (6)), but uses inverse conditional CDFs $C_{d|1,\ldots,d-1}^{-1}$ for each $V_d$. We use the notation $C^{-1}$ to emphasise that in the copula flow case we are dealing with inverse *copula* CDFs, whose codomain is also uniform.

Autoregressive flows estimate each univariate conditional flow $\mathcal{C}_{d|1\ldots d-1}$ with a strictly monotone function whose parameters are only allowed to depend on dimensions $1, \ldots, d-1$. The monotonicity of the function ensures invertibility, while the autoregressive structure in the function parameter dependence gives a triangular Jacobian whose determinant is tractable. Kamthe et al. (2021) use a NSF, where the monotone function is given by a monotone rational quadratic spline, whose knot parameters are provided by a neural network where weights are appropriately masked to ensure the autoregressive structure. The univariate marginal flows $\mathcal{F}_{\boldsymbol{X}}$ are estimated with separate NSFs before training the copula flow using the transformed data $\boldsymbol{V}$.

While a NSF can constrain the support of both the base and target distributions, it cannot control the form of the marginal distribution. If marginal and copula flows are learned simultaneously, neither will be correctly inferred due to the infinite possible combinations of $(\mathcal{F}_{\boldsymbol{X}}, \mathcal{C}_{\boldsymbol{X}})$ which yield the same composite flow $\mathcal{W} = \mathcal{F}_{\boldsymbol{X}} \circ \mathcal{C}_{\boldsymbol{X}}$. These flows must be learned sequentially if $\mathcal{C}$ is to model a copula.

In our application, we wish to infer a multivariate copula which models the joint dependence between univariate pretreatment covariates and conditional univariate quantities such that the latter parameterises the causal margin. Inferring the MOD separately from the copula, as copula-based flows do, will target the conditional causal effect rather than the marginal causal effect. We propose a solution in the form of Frugal Flows, which we introduce in Section 3.1.1. Moreover, for discrete variables we use a dequantised form of the empirical CDF rather than a NSF adaptation (see Appendix B.2 for further details).

## 2.6 Validating and Benchmarking Causal Methods

Methods for validating causal models can be broadly categorised into two groups. The first comprises auxiliary analyses conducted after fitting a causal model and estimating a treatment effect. These include but are not limited to sensitivity analyses (Imai et al., 2010), subgroup analyses (Cochran and Chambers, 1965), placebo tests (Eggers et al., 2023), and negative controls (Shi et al., 2020).

The second set of validation methods is where we see FFs having a significant impact. These methods are used to construct synthetic datasets while allowing the causal practitioner to customise specific features of the data-generating process. For example, when validating an inference method which estimates an ATE under certain confounding assumptions, it is crucial that generated data follow the "ground truth" ATE and confounding assumptions one wishes to measure. However, synthetic data risk being oversimplified and contrived, failing to reflect the complexity of real world datasets.

To mitigate this, generative models are trained on real-world data and calibrated to generate samples with modifiable causal constraints. Such constraints include the average causal treatment effect, unobserved confounding, and positivity. To our knowledge, the FF framework proposed in this paper is the first method to allow all of these conditions to adjusted by the user. Existing methods (Neal et al., 2020; Athey et al., 2021; Parikh et al., 2022) encode these effects through soft optimisation constraints, hence there is no guarantee that the constraints are satisfied. Enforcing these constraints too strongly may negatively impact model optimisation, and may affect the reconstructive ability of the underlying model. Furthermore, since these approaches do not explicitly parameterise the causal effect, samples from trained models must be tested *post hoc* to ensure the desired constraints are present in the sampled data. A key benefit of frugal models is that the marginal causal effect is directly parameterised by the user through the likelihood. As a result, synthetic data samples will exactly satisfy these constraints.

# 3 Method

## 3.1 Building the Joint Distribution

In this section we parameterise the full observational joint using FFs. Section 3.1.1 outlines how the FF is constructed; we first learn the probability integral transforms of the pretreatment covariates, and then infer the causal margin jointly with an extended copula flow, the Frugal Flow. To infer the causal margin, this is sufficient. Nevertheless, the propensity score is needed to complete the joint in order to generate benchmarks which are confounded in a similar fashion to the original real-world dataset. We describe the fitting of the propensity score in Section 3.1.2

### 3.1.1 Constructing Frugal Flows

The first step involves learning the margins for the pretreatment covariates $\boldsymbol{Z}$. This is done in a similar fashion to that of Kamthe et al. (2021)'s copula-based flows, as described in Section 2.5. The outcome, treatment, and the inferred ranks $\boldsymbol{V_Z}$ of the pretreatment covariates are then used to train the Frugal Flow (see bottom part of Figure 2) that models $\mathcal{F}_{Y|\mathrm{do}(T)}^{-1}$ together with the copula flow. This is required in order to learn the causal marginal $p_{Y|\mathrm{do}(T)}$ rather than the conditional $p_{Y|T}$.

The Frugal Flow of dimension $D+1$ transforms the joint input of $(Y, \boldsymbol{V_Z} \mid \mathrm{do}(T))$ into a random vector $\boldsymbol{U}$ which we set to be distributed according to an independent uniform base distribution. In the first subflow of the composition, $Y$ is pushed through a univariate flow $\mathcal{F}_{Y|\mathrm{do}(T)}^{-1}$ conditioned on $T$ to obtain $V_{Y|\mathrm{do}(T)}$, while the $\boldsymbol{V_Z}$ remain untransformed. Subsequently, $V_{Y|\mathrm{do}(T)}$ is kept fixed, while a copula is learnt over $\boldsymbol{V_Z}$ conditional on $V_{Y|\mathrm{do}(T)}$ via an NSF. Importantly, a specific ordering of the variables is imposed, such that the causal margin is ranked first. In this way, we ensure that $U_1$ and $V_{Y|\mathrm{do}(T)}$ have the same distribution, and $V_{Y|\mathrm{do}(T)}$ is therefore constrained to be uniform. The marginal flow $\mathcal{F}_{Y|\mathrm{do}(T)}^{-1}$ thus targets the CDF of the marginal causal effect, $F_{Y|\mathrm{do}(T)}$.

In summary, we construct a flow $\mathcal{Q}^{-1} : (Y, V_{Z_1}, \ldots, V_{Z_D} \mid T) \to \boldsymbol{V}$ as a composition of a marginal flow $\mathcal{F}_{Y|\mathrm{do}(T)}^{-1}$ and conditional copula distribution $\mathcal{C}^{-1}(v_{Y|\mathrm{do}(T)}, v_{Z_1}, \ldots, v_{Z_D}) = C(v_{Z_1}, \ldots, v_{Z_D} \mid v_{Y|\mathrm{do}(T)})$. More on the implementation details can be found in Appendix C.

### 3.1.2 Learning the Propensity Flow

We constructed the conditional distribution of $Y$ and $\boldsymbol{Z}$ after an intervention on $T$ in Section 3.1.1:

$$p_{\boldsymbol{Z}Y|\mathrm{do}(T)}(\boldsymbol{z}, y \mid t) = \left[\prod_{i=1}^{D} p_{Z_i}(z_i)\right] \cdot p_{Y|\mathrm{do}(T)}(y, t) \cdot c_{\boldsymbol{Z}Y|\mathrm{do}(T)}(v_{Y|\mathrm{do}(T)}, v_{Z_1}, \ldots, v_{Z_D}).$$

$$\begin{bmatrix} Z_1 \\ \vdots \\ Z_D \end{bmatrix} \rightarrow \begin{pmatrix} \mathcal{F}_{Z_1}^{-1}(\cdot) \\ \vdots \\ \mathcal{F}_{Z_D}^{-1}(\cdot) \end{pmatrix} \rightarrow \begin{bmatrix} V_{Z_1} \\ \vdots \\ V_{Z_D} \end{bmatrix} = \boldsymbol{V_Z}$$

$$\begin{bmatrix} Y \mid \mathrm{do}(T) \\ \boldsymbol{V_Z} \end{bmatrix} \rightarrow \begin{pmatrix} \mathcal{F}_{Y\mid\mathrm{do}(T)}^{-1}(\cdot) \\ \mathbb{I}(\cdot) \end{pmatrix} \rightarrow \begin{bmatrix} V_{Y\mid\mathrm{do}(T)} \\ \boldsymbol{V_Z} \end{bmatrix} \rightarrow \mathrm{NSF}\begin{pmatrix} c(v_{Y\mid\mathrm{do}(T)}) = 1 \\ c(\boldsymbol{v_Z} \mid v_{Y\mid\mathrm{do}(T)}) \end{pmatrix} \rightarrow \begin{bmatrix} U_1 \\ \boldsymbol{U}_{2:(D+1)} \end{bmatrix}$$

Figure 2: Structure for learning a Frugal Flow. The top line outlines the process for learning the univariate marginal flows of the pretreatment covariates $\boldsymbol{Z}$. The bottom transform illustrates the Frugal Flow, which learns the conditional copula $c(\boldsymbol{v_Z} \mid v_{Y\mid\mathrm{do}(T)})$ jointly with the causal marginal flow $\mathcal{F}_{Y\mid\mathrm{do}(T)}$ by enforcing $V_{Y\mid\mathrm{do}(T)}$ to be marginally uniform.

Inferring the above is sufficient for identifying the causal margin. However, to generate realistic samples for causal method validation, one also needs to learn the propensity score, $p_{T\mid\boldsymbol{Z}} = p_T \cdot c_{T\mid\boldsymbol{Z}}$. By decoupling the marginal treatment density $p_T$ from the conditional copula $c_{T\mid\boldsymbol{Z}}$, one can can modify the marginal treatments while retaining the dependence of the original data. We therefore learn an approximate probability integral transform of the discrete treatment $T$ (see Appendix B.2.1 for further details), followed by the conditional copula flow of $T$ on $\boldsymbol{Z}$, $\mathcal{C}_{T\mid\boldsymbol{Z}}^{-1} : V_T \rightarrow V_{T\mid\boldsymbol{Z}} \mid \boldsymbol{Z}$.

One could directly model $p_{T\mid\boldsymbol{Z}}$ using a normalising flow, which would also constitute a valid frugal model. We instead choose to model the conditional copula using a flow, $C_{T\mid\boldsymbol{Z}} = \mathcal{C}_{T\mid\boldsymbol{Z}}^{-1}$, allowing users to encode a degree of unobserved confounding in the generated data by sampling the ranks $V_{T\mid\boldsymbol{Z}}$ and $V_{Y\mid\mathrm{do}(T)}$ from a non-independence copula. Assuming ignorability, these ranks would be independent. However, unobserved confounders imply dependence between these ranks. Sampling them from a copula can replicate this effect, as demonstrated in the far-right plots in Figures 3 and 4.

The above section describes how one can estimate the propensity of treatment from a real-world dataset. However, we remark that one can choose any custom propensity score function to generate treatments conditional on the pretreatment covariates via inverse probability integral transforms on $V_{T\mid\boldsymbol{Z}}$. Hence, one can fully control the overlap/positivity of FF generated benchmark datasets.

## 3.2 Generating Synthetic Benchmarks

Data generated from a fitted FF can be customised with a range of properties, allowing for model validation against a range of customisable causal assumptions. We describe these below.

**Modifying the Causal Margin**  The central output of the Frugal Flow is a method for sampling ranks for each of the margins in $P_{Y\mid\mathrm{do}(T)}, P_{Z_1}, \ldots, P_{Z_d}$. Any causal marginal density $q_{Y\mid\mathrm{do}(T)}$ can be used to generate samples of $Y$ via inverse probability integral transforms. Since the Frugal Flow returns ranks for the intervened causal effect, these can be inverse transformed by any valid CDF. Unlike other methods, this constraint is strictly enforced by the the frugal likelihood.

**Simulating from Discrete Outcomes**  Since FFs return $V_{Y\mid\mathrm{do}(T)}$ ranks, one can sample from any custom causal margin. This extends to both continuous and discrete causal margins. One can simulate from a logistic marginal effect $Y \mid \mathrm{do}(T) \sim \mathrm{Bernoulli}(p = \mathrm{expit}(\beta T + c))$ or probit model $Y \mid \mathrm{do}(T) \sim \mathrm{Bernoulli}(p = \Phi(\beta T + c))$ where $\Phi(\cdot)$ is a univariate standard Gaussian CDF. This is non-trivial, because logistic regression is not *collapsible*, meaning that if (for example) $Y \mid T = t, \boldsymbol{Z} = \boldsymbol{z}$ is a logistic regression, then $Y \mid \mathrm{do}(T = t)$ generally will not be. Hence it is infeasible for a fully conditional method of simulation to produce outcomes where the causal margin uses a logistic link. For experimental results see Appendix D.2.1.

**Modifying the Degree of Unobserved Confounding**  One can sample data from FFs as if the outcome is affected by unobserved confounding. The variables $V_{Y\mid\mathrm{do}(T)}$ and $V_{T\mid\boldsymbol{Z}}$ are independent of each other if no unobserved confounding is assumed. Introducing a dependence between these ranks replicates the effect of unobserved confounding. This can be achieved by sampling $(V_{Y\mid\mathrm{do}(T)}, V_{T\mid\boldsymbol{Z}})$ from a Gaussian bivariate copula, $c(v_{Y\mid\mathrm{do}(T)}, v_{T\mid\boldsymbol{Z}}; \rho)$, where $\rho$ quantifies the degree of unobserved confounding in the sampled data.

**Customising Treatment Effect Heterogeneity**  Consider a stationary treatment with pretreament covariate set $\boldsymbol{Z} = (\boldsymbol{W}, \overline{\boldsymbol{W}})$ where $\boldsymbol{W} \subset \boldsymbol{Z}$ with $|\boldsymbol{Z}| = D$, $|\boldsymbol{W}| = d$, and $|\overline{\boldsymbol{W}}| = D - d$. We proceed considering the case where $0 < d < D$. Interest may lie in the causal treatment margin **conditional** on the subset of variables $\boldsymbol{W}$:

$$p_{Y|\boldsymbol{W},\mathrm{do}(T)}(y \mid \boldsymbol{w}, t) = \int_{\overline{\mathcal{W}}} d\overline{\boldsymbol{w}} \, p_{Y|\boldsymbol{Z},\mathrm{do}(T)}(y \mid \boldsymbol{w}, \overline{\boldsymbol{w}}, t) \, p_{\overline{\boldsymbol{W}}|\boldsymbol{W}}(\overline{\boldsymbol{w}} \mid \boldsymbol{w}) \tag{7}$$

We propose a method to exactly parameterise heterogeneous treatment effects using a subset of pretreatment covariates, $\boldsymbol{W} \subset \boldsymbol{Z}$. FFs offer exact parameterisation of $p_{Y|\boldsymbol{W},\mathrm{do}(T)}$, allowing for customisation of heterogeneity while capturing complex dependencies between other covariates. Specifically, the model infers the conditional treatment margin, $p_{Y|\boldsymbol{W},\mathrm{do}(T)}(y \mid \boldsymbol{w}, t)$, ensuring proper inference of the joint pretreatment covariate distribution, $p_{\boldsymbol{Z}}(\cdot)$. Thus, one may simulate data where causal effects are conditional on a selected subset of variables, offering flexible and precise control over treatment heterogeneity. Further details may be found in Appendix D.2.2.

**Customising the Propensity Score**  Since the propensity score is variation independent from the rest of the model, one has complete flexibility on how to parameterise the propensity score. Any distribution $P_{T|\boldsymbol{Z}}$ can be used to generate treatments with varying degrees of overlap in a manner that is completely customisable by the user.

## 4  Experiments

The following section discusses our experiments and results, which aim to i) demonstrate that FFs accurately infer the true MOD for confounded data, and ii) show how a trained FF can generate synthetic datasets that meet user specified causal margins and unobserved confounding.

### 4.1  Inference

We generate simulated data from three models. The first two are parameterised by four pretreatment covariates $\boldsymbol{Z} = \{Z_1, \ldots, Z_4\}$ with a binary treatment $T$, a linear Gaussian causal margin $Y \mid \mathrm{do}(T) \sim \mathcal{N}(\mu = T + 1, \sigma = 1)$, and a copula dependence measure $c(v_{Y|T}, v_{Z_1}, \ldots, v_{Z_4})$. In the first model $M_1$, all four covariates follow a gamma distribution. In the second $M_2$, the data is generated from an even split of gamma and binary covariates. Additionally, we generate data from model $M_3$ with ten pretreatment covariates comprising five gamma and five binary variables. A more quantitative description of the simulated data generating process and hyperparameter values are presented in Appendix D.1.

Table 1: Mean and $2\sigma$ confidence interval of the inferred ATE, bootstrapped over 25 different runs and with a data size of $N = 25{,}000$. The number of pretreatment covariates in each model is denoted by $D$. Bold confidence intervals contain the true ATE. OR quotes the results obtained by linear outcome regression, and CNF reports the ATE estimted by causal normalising flows.

| Model | True ATE | $D$ | Frugal Flow | OR | Matching | CNF |
|-------|----------|-----|-------------|-----|----------|-----|
| $M_1$ | 1 | 4 | $\mathbf{0.98 \pm 0.12}$ | $1.28 \pm 0.06$ | $\mathbf{0.78 \pm 1.06}$ | $0.73 \pm 0.16$ |
| $M_1$ | 5 | 4 | $\mathbf{5.00 \pm 0.24}$ | $5.29 \pm 0.04$ | $\mathbf{4.68 \pm 1.06}$ | $4.23 \pm 0.20$ |
| $M_2$ | 1 | 4 | $\mathbf{1.01 \pm 0.10}$ | $1.46 \pm 0.07$ | $\mathbf{1.36 \pm 0.72}$ | $\mathbf{1.01 \pm 0.20}$ |
| $M_2$ | 5 | 4 | $\mathbf{5.01 \pm 0.18}$ | $5.44 \pm 0.05$ | $\mathbf{5.55 \pm 0.88}$ | $\mathbf{5.03 \pm 0.44}$ |
| $M_3$ | 1 | 10 | $\mathbf{1.00 \pm 0.09}$ | $1.13 \pm 0.06$ | $\mathbf{0.90 \pm 0.48}$ | $0.87 \pm 0.15$ |
| $M_3$ | 5 | 10 | $\mathbf{5.18 \pm 0.30}$ | $5.13 \pm 0.26$ | $\mathbf{4.90 \pm 0.47}$ | $4.73 \pm 0.28$ |

We generated datasets with a sample size of $N = 25{,}000$ across $B = 25$ different runs. Frugal Flows (FFs) were compared against outcome regression (OR), traditional causal propensity score matching (Stuart, 2010), and state-of-the-art causal normalising flows (CNFs) (Javaloy et al., 2024). Further details on the methods can be found in Appendix D.3.3. A default set of hyperparameters was used for all models. The estimated ATEs are shown in Table 4. OR models, which estimate the *conditional* rather than the *marginal* effect of $T$ on $Y$, consistently exhibited bias, pulling the

estimates away from the true value. In contrast, FFs achieved the lowest error in identifying the true ATE, outperforming both statistical matching and CNFs.

Our results demonstrate that Frugal Flows can correctly identify causal relationships under ideal conditions, confirming that they are a valid, efficient way to parameterise a causal model using deep learning architectures. A drawback of FFs is that they need large datasets to accurately infer causal margins. Additionally, the complexity of data dependencies might require careful hyperparameter tuning to prevent the copula from overfitting, which could bias the inference of the causal relationships. Because of these challenges, we do not recommend using FFs on real-world datasets for statistically inferring treatment effect sizes, as causal benchmark datasets are usually small.

## 4.2 Benchmarking and Validation

In this section we present the results of multiple causal inference methods on data generated from FFs trained on two real-world datasets. The first is the Lalonde data, taken from a randomised control trial to study the effect of a temporary employment program in the US on post intervention income level (LaLonde, 1986). The second is an observational dataset used to quantify the effect of individuals' 401(k) eligibility on their accumulated net assets, in the presence of several relevant covariates (Abadie, 2003). Both datasets have a binary treatment and continuous outcome. Appendix D.3 can be referred to for a more comprehensive description of the data. In addition, we present diagnostics on the quality of the model fit in Appendix D.3.6, and the loss optimization for both datasets is presented in Appendix D.3.7.

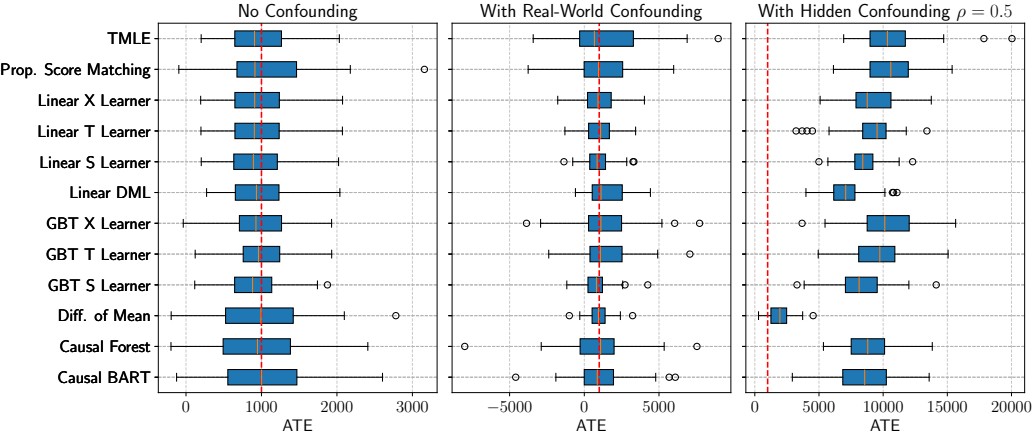

Figure 3: Boxplot of ATE estimates from 10 inference methods, estimated across 50 different samples from a FF trained on the Lalonde dataset. The dotted red line represents the customized ATE of samples generated from the trained Frugal Flow.

FFs were fitted to both datasets and used to simulate data with an ATE of $1000$. We simulate 50 datasets of size $N = 1000$ from three different cases each: i) no confounding, ii) with confounding according to the propensity flow inferred in the model fitting, and iii) with propensity flow confounding **and** unobserved confounding introduced via a Gaussian copula. A variety of causal inference methods (see Appendix D.3.4 for a more detailed description) were fit to the data sets, including a difference of means (DoMs) estimate which is an unbiased estimator of the treatment effect for randomised data. The inferred ATEs across all runs are presented in Figure 3 and Figure 4.

In both cases, all inference methods demonstrate no bias when fitted to unconfounded data. With real-world confounding, most methods estimate the correct ATE in Figure 4, whereas the DoMs shows a substantial bias from the ground truth. In Figure 3 however, all methods infer the correct ATE including DoMs. This is not surprising as the original data was randomised; the propensity flow here appears to simply add more noise to the outcomes. Finally, we note that all causal inference methods show confounding bias in the far right hand plots, demonstrating that FFs can simulate data with replicate the effects of hidden confounding.

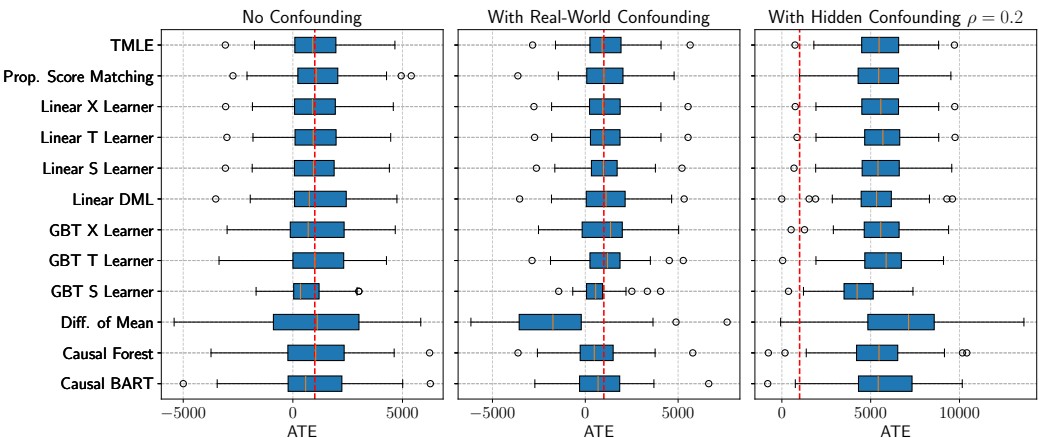

Figure 4: Boxplot of ATE estimates from 10 inference methods, estimated across 50 different samples from a FF trained on the e401(k) dataset. The dotted red line represents the customized ATE of samples generated from the trained Frugal Flow.

## 5 Conclusions

We introduce Frugal Flows, a novel likelihood-based model that leverages NFs to flexibly learn the data-generating process while directly targeting the marginal causal quantities inferred from observational data. Our proposed model addresses the limitations of existing methods by expliclitly parameterising the causal margin. FFs offer significant improvements in generating benchmark datasets for validating causal methods, particularly in scenarios with customizable degrees of unobserved confounding. To our knowledge, FFs are the first generative model that allows for exact parameterisation of causal margins, including binary outcomes from logistic and probit margins.

### 5.1 Limitations and Future Work

Our experiments validated the empirical effectiveness of FFs, showing that they can infer the correct form of causal margins on confounded data simulations. Despite these promising results, FFs come with certain limitations that need to be addressed in future research. NFs require extensive hyperparameter tuning, which can be computationally intensive and time-consuming. Moreover, we see that FFs perform better in inference tasks with larger datasets. Future work could explore alternative ML copula methods and architectures that may be more effective for smaller datasets. Fortunately, this is less problematic for simulation as specification of the exact causal margin is left to the user. Additionally, the dequantising mechanism used by FFs implicitly shuffles the order of discrete samples, potentially losing some inherent structure in the data, making FFs less suitable for categorical datasets without implicit ordering.

In summary, Frugal Flows offer a novel approach to causal inference and model validation that combines flexibility with exact parameterisation of causal effects. Future work will refine the inference capabilities and extend the applicability of FFs to a wider range of data types and sizes.

## Acknowledgements

The authors express their deep gratitude to Stefano Cortinovis and Silvia Sapora for their valuable suggestions regarding the development of the software and experiments. We also thank Christopher Williams for his insightful advice on the framing of the paper. We thank Geoff Nicholls for his suggestions and fruitful discussions on the paper and opportunities for future development. Last but certainly not least, special thanks must go to Shahine Bouabid for his invaluable assistance with both the coding aspects of this paper and his recommendations on clarity.

DdVM is supported by a studentship from the UK's Engineering and Physical Sciences Research Council's Doctoral Training Partnership (EP/T517811/1). LB is supported by a Clarendon Scholarship.

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

# A The Frugal Parameterisation

The frugal parameterisation proposed by Evans and Didelez (2024) provides a method for simulating from a parametric marginal causal model, by starting with this distribution and building the rest of the model around it.

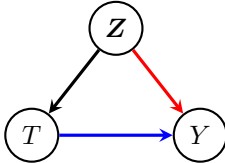

Figure 5: A generalised example of a static causal treatment model. The past $P(T, \boldsymbol{Z})$ (black) can be freely specified separately from the causal effect (blue). However, the dependency measure between $\boldsymbol{Z}$ and $Y$ (red), $\phi$ should be parameterised in such a way that the margins $P(\boldsymbol{Z})$ and $P(Y | \mathrm{do}(T))$ are invariant to changes in $\phi$.

We specify the notation used in this appendix. Functions labelled $F_i(\cdot)$ are CDFs for the variable $i$. Apart from this, in general density functions will be labelled with a lower case letter, whereas CDFs will be named with the upper case (e.g. we contrast the copula density $c(u_1, u_2)$ with the distribution function $C(u_1, u_2)$).

Consider firstly the case of a static treatment model with a single outcome $Y$, a single treatment $T$ and an effective pretreatment covariate set $\boldsymbol{Z}$. Assume that any of these covariates occur prior to treatment even if they do not causally affect the treatment directly. Evans and Didelez (2024) construct frugal models in three parts:

- The causal distribution of interest $P(Y \mid \mathrm{do}(T))$
- The past $P(\boldsymbol{Z}, T)$
- The intervened variation independent dependency measure $\phi(Y, \boldsymbol{Z} \mid \mathrm{do}(T))$.

The three frugal components are *variation independent* in the sense that they characterise non-overlapping components of the full observational joint. We quote the following definition from Evans and Didelez (2024):

**Definition 1** *Take a set $\Theta$ and two functions defined on it $\phi, \psi$. We say that $\phi$ and $\psi$ are variation independent if $(\phi \times \psi)(\Theta) = \phi(\Theta) \times \psi(\Theta)$; i.e. the range of the pair of functions together is equal to the Cartesian product of the range of them individually.*

Variation independence (VI) is a highly desirable property for a parameterization, since it allows different components to be specified entirely separately. This is extremely useful if one is trying to use a link function in a GLM, or to specify independent priors for a Bayesian analysis. In addition, VI is important in semiparametric statistics. The definition simply states that the Cartesian product of the images is the same as the image of the joint map. For example, in a bivariate gamma-distribution with positive responses, then $\mu_1 \in \mathbb{R}^+$ and $\mu_2 \in \mathbb{R}^+$ is a variation independent parameterization, since

$$(\mu_1 \times \mu_2)(\Theta) = \mathbb{R}^+ \times \mathbb{R}^+ = \mu_1(\Theta) \times \mu_2(\Theta).$$

However, if we replace $\mu_2$ with $\mu_2' = \mu_2 - \mu_1$ (for example), then although the range of this parameter is $\mathbb{R}$,

$$(\mu_1 \times \mu_2')(\Theta) = \{(x, y) : x > 0, y > -x\} \neq \mathbb{R}^+ \times \mathbb{R} = \mu_1(\Theta) \times \mu_2'(\Theta).$$

Central to this is the choice of $\phi^*$. This dependency measure should encode dependencies between $\boldsymbol{Z}$ and $Y \mid \mathrm{do}(T)$, but not provide information about their marginal distributions.

Discrete frugal models can be parameterised by conditional odds ratios, while continuous variables typically use copulae. Both allow for variation independent parameterisation of the full joint distribution. The methodology facilitates the creation and simulation of models with parametrically determined causal distributions, enabling fitting using likelihood-based techniques, including fully Bayesian methods. Furthermore, this parameterisation covers a range of causal quantities, such as the average causal effect and the effect of treatment on the treated.

# B Copula Theory

Copulae present a powerful tool to model joint dependencies independent of the univariate margins. This aligns well with the requirements of the frugal parameterisation, where dependencies need to be varied without altering specified margins (the most critical being the specified causal effect). Understanding the constraints and limitations of copula models ensures that causal models remain accurate and consistent with the intended parameterisation.

## B.1 Sklar's Theorem

Sklar's theorem (Sklar, 1959; Czado, 2019) is the fundamental foundation for copula modelling, as it provides a bridge between multivariate joint distributions and their univariate margins. It allows one to separate the marginal behaviour of each variable from their joint dependence structure, with the latter being represented by the copula itself.

**Theorem 1** *For a d-variate distribution function $F_{1:d} \in \mathcal{F}(F_1, \ldots, F_d)$, with $j^{th}$ univariate margin $F_j$, the copula associated with $F$ is a distribution function $C : [0,1]^d \to [0,1]$ with uniform margins on $(0,1)$ that satisfies*

$$F_{1:d}(\boldsymbol{y}) = C(F_1(y_1), \ldots, F_d(y_d)), \boldsymbol{y} \in \mathbf{R}^d.$$

*1. If $F$ is a continuous d-variate distribution function with univariate margins $F_1, \ldots, F_d$ and rank functions $F_1^{-1}, \ldots, F_d^{-1}$ then*

$$C(\boldsymbol{u}) = F_{1:d}(F_1^{-1}(u_1), \ldots, F_d^{-1}(u_d)), \boldsymbol{u} \in [0,1]^d.$$

*2. If $F_{1:d}$ is a d-variate distribution function of discrete random variables (more generally, partly continuous and partly discrete), then the copula is unique only on the set*

$$Range(F_1) \times \cdots \times Range(F_d).$$

*The copula distribution is associated with its density $c(\cdot)$*

$$f(\boldsymbol{y}) = c(F_1(y_1), \ldots, F_d(y_d)) \cdot f_1(y_1) \ldots f_d(y_d)$$

*where $f_i(\cdot)$ is the univariate density function of the $i^{th}$ variable.*

Note that Sklar's theorem explicitly refers to the **univariate marginals** of the variable set $\{Y_1, \ldots, Y_d\}$ to convert between the joint of univariate margins $C(\boldsymbol{u})$ and the original distribution $F(\boldsymbol{y})$. For absolutely continuous random variables, the copula function $C$ is unique. This uniqueness no longer holds for discrete variables, but this does not severely limit the applicability of copulae to simulating from discrete distributions. The non-uniqueness does play a more problematic role in copula inference, however (Genest and Nevslehova, 2007).

An equivalent definition (from an analytical purview) is $C : [0,1]^d \to [0,1]$ is a $d$-dimensional copula if it has the following properties:

1. $C(u_1, \ldots, 0, \ldots, u_d) = 0$
2. $C(1, \ldots, 1, u_i, 1, \ldots, 1) = u_i$.
3. $C$ is $d$-non-decreasing.

**Definition 2** *A copula $C$ is d-non-decreasing if, for any hyperrectangle $H = \prod_{i=1}^d [u_i, y_i] \subseteq [0,1]^d$, the C-volume of $H$ is non-negative.*

$$\int_H C(\boldsymbol{u}) \, d\boldsymbol{u} \geq 0$$

## B.2 Copulae for Discrete Variables

Accurately modelling the univariate marginal CDFs of pretreatment covariates is a crucial step in training Frugal Flows, particularly when the dataset includes discrete variables. For continuous covariates, the mapping between observations and ranks is unique, allowing for straightforward

estimation of the marginal distribution. However, with discrete covariates, this mapping becomes one-to-many, as the same observation can be transformed from different ranks. This non-uniqueness introduces significant challenges when modelling the joint distribution via copulae, as the joint set of ranks for discrete covariates lacks a unique representation. As a result, estimating the dependency structure between variables becomes more complex and less reliable.

To extend Frugal Flows to accommodate mixed data types, it is essential to generate empirical ranks for discrete covariates in a way that allows the model to capture their dependencies. Our goal is to obtain valid rank samples that can be used to train a Frugal Flow without introducing distortions in the copula structure. While this issue has been widely explored in the copula literature for parametric models, there remains a gap in effectively addressing it within more flexible, non-parametric frameworks. In this work, we implement a generalised distributional transform for discrete covariates (presented in Appendix B.2), which allows Frugal Flows to be trained effectively, maintaining the flexibility of the model while accurately capturing the relationships between variables.

### B.2.1 Challenges and Motivation

In addition to the above, copulae encode a degree of ordering in the joint as probability integral transforms are inherently ranked, and hence should only be used for variables that have an inherent ordering of their own (e.g. count or ordinal data models). While approaches to model discrete variables exist in parametric copulae models (Zilko and Kurowicka, 2016; Panagiotelis et al., 2017), more flexible non-parametric copulae struggle to capture the dependencies of empirical copulae. Similar to Kamthe et al. (2021) we use the approach suggested by Rüschendorf (2009). An outline of this method is presented in Appendix B.2.2. However, unlike Kamthe et al. we use the empirical CDF inferred from the discrete data as opposed to modelling the CDF with a marginal flow.

### B.2.2 Empirical Copula Processes for Discrete Variables

In order to deal with discrete variables, we use a similar approach as taken by Kamthe et al. (2021), who quote the generalised distributional transform of a random variable found originally proposed by Rüschendorf (2009). We quote the main result from Rüschendorf (2009) below.

**Theorem 2** *On a probability space* $(\Omega, \mathcal{A}, P)$ *let* $X$ *be a real random variable with distribution function* $F$ *and let* $V \sim U(0, 1)$ *be uniformly distributed on* $(0, 1)$ *and independent of* $X$*. The modified distribution function* $F(x, \lambda)$ *is defined by*

$$F(x, \lambda) := P(X < x) + \lambda P(X = x).$$

*We define the (generalised) distributional transform of* $X$ *by*

$$U := F(X, V).$$

*An equivalent representation of the distributional transform is*

$$U = F(X-) + V(F(X) - F(X-)).$$

Rüschendorf (2009) makes a key remark about the generalised transform's lack of uniqueness for discrete variables. Such a dequantisation step may introduce artificial local dependence which may lead to an incorrect flow being inferred, and therefore hinder the inference of the causal margin.

## C   Frugal Flow Implementation Details

The FF software used for this paper can be found in the GitHub repository https://github.com/llaurabatt/frugal-flows.git.

FF software builds upon FlowJax (Ward, 2024), a Python package implementing normalising flows in JAX (Bradbury et al., 2018). JAX is an open-source numerical computing library that extends NumPy functionality with automatic differentiation and GPU/TPU support, designed for high-performance machine learning research.

**Frugal Flow architecture.**

The Frugal Flow component builds a flow of the form in the bottom part of Figure 2. It allows us to implement $\mathcal{F}_{Y|T}$ with either (i) a customised CDF conditioned on $T$, of a known parametric family;

(ii) a univariate NSF on the $[-1, 1]$ interval modified to allow a location translation parameter that represents the ATE for $T$, where the input is mapped from the real line via a tanh transform; or finally (iii) a univariate NSF on the $[-1, 1]$ interval that is not conditional on $T$, where the input is mapped from $[0, 1]$ via an affine transform. As for known parametric families, only the Gaussian CDF is currently implemented, but the architecture allows us to define any different parametric class provided that it constitutes a diffeomorphism. As for the univariate NSF in (iii), it does not explicitly learn an ATE in the training phase, but can be used for simulation of e.g. binary outcomes by applying a subsequent logistic transformation to its outcome.

The multivariate NSF element is a composition of multiple modified NSF subflows. In each subflow the first transform is fixed to be an identity, while the other dimensions are transformed with a monotone rational quadratic spline whose knot parameters are produced by masked multilayer perceptrons (MLPs) implemented as in Germain et al. (2015), conditional on the first dimension. In order to increase expressivity, dimension permutation is usually applied between the different subflows in the NSF composition. We still allow this permutation but excluding the first dimension, that is fixed to be at the top in each subflow. The NSF acts on the on the $[-1, 1]$ interval and is mapped from and back to the quantile space with affine transforms.

Tunable hyperparameters to the Frugal Flow component are the number of subflows of the multivariate NSF, the width and depth of the MLPs and the number of spline knots, together with the specific hyperparameters of the chosen $\mathcal{F}_{Y|T}$.

**Marginal flows architecture for continuous variables.**

Each marginal flow for the continuous covariates maps each variable from the real line to the $[-1, 1]$ interval with a tanh transform, then applies a univariate NSF on the $[-1, 1]$ interval, and maps back to the standard uniform base distribution via an affine transform. Tunable hyperparameters are the number of subflows of the NSF, the width and depth of the MLPs and the number of spline knots.

**Marginal transform architecture for discrete variables.**

To map a discrete $Z$ to the ranks $V_Z$, we compute its empirical CDF and then apply the procedure outlined in Appendix B.2.2. We use the inverse of the same empirical CDF to map ranks back to the $Z$ for sampling.

**Propensity score model architecture.**

To map a discrete $T$ to the ranks $V_T$, we compute its empirical CDF and then apply the procedure outlined in Appendix B.2.2. A univariate NSF flow with a uniform base distribution is then applied to learn the copula CDF of $T$ on the $[0, 1]$ support, conditioned on $\mathbf{Z}$. The $\mathbf{Z}$ conditioning is obtained by adding $\mathbf{Z}$ as an (unmasked) input to the MLP that produces the knot parameters for the rational quadratic spline. This is standard in NF literature. Tunable hyperparameters are the number of subflows of the NSF, the width and depth of the MLPs and the number of spline knots.

The propensity score model can be inverted to generate $T$ samples conditioned on a given $\mathbf{Z}$. Uniform samples are pushed through the trained univariate propensity score flow to obtain ranks, that are then mapped to the discrete space via the inverse of the empirical CDF of $T$.

**Training the Frugal Flow and the propensity score flow.**

In order to train a FF, one must fit the marginal flows first. The marginal flows are trained (for continuous variables) via maximising the log-likelihood with stochastic gradient descent, and/or the discrete $\mathbf{Z}$ are mapped to the rank space via the procedure in Appendix B.2.2. Next, the FF is trained via maximum likelihood estimation (MLE), taking as input the outcome $Y$ together with the ranks $V_Z$, and conditioning the flow for the causal margin on treatment $T$ where required by the chosen method. For MLE optimisation, we take advantage of JAX automatic differentiation capabilities and use the Adam optimiser (Kingma and Ba, 2015), whose hyperparameters can also be tuned. If required, the propensity flow is likewise trained on $V_T$ conditioning on $\mathbf{Z}$ via MLE with an Adam optimiser.

**Simulating benchmarks.**

One can use a trained FF for simulation of causal benchmarks. The general data simulation pipeline is:

1. Generate a sample of $U_{T|\boldsymbol{Z}}, V_{Y|\operatorname{do}(T)}$ from a bivariate Gaussian copula, parameterised by correlation $\rho$, which quantifies the degree of unobserved confounding one wishes to encode in the benchmark. If no unobserved confounding is desired, set $\rho = 0$.

2. Generate a sample of $D$ independent uniforms, $\boldsymbol{U_Z}$

3. Push the sample $(V_{Y|\operatorname{do}(T)}, \boldsymbol{U_Z})$ through the trained FF and save the resulting correlated $\boldsymbol{V_Z}$ samples associated with $V_{Y|\operatorname{do}(T)}$

4. Generate $\boldsymbol{Z}$ from uniform samples via inverting the univariate marginal flows (continuous variables) and/or using the learnt inverse empirical CDF (discrete variables)

5. Generate $T$ as a function of $\boldsymbol{Z}$ by pushing $U_{T|\boldsymbol{Z}}$ through the inverse of the trained propensity score flow and mapping ranks $V_T$ back to the discrete space via the learnt inverse empirical CDF

6. Push $V_{Y|\operatorname{do}(T)}$ through the desired causal margin transform to obtain outcome samples conditioned on $T$. Currently, the package supports:

   (a) Sampling from an inverse CDF provided by the user, taking $V_{Y|\operatorname{do}(T)}$ as input and conditioning on the given univariate $T$. Currently a Gaussian inverse CDF is implemented, where the coefficient on $T$ can be chosen to impose the desired ATE. The user is free to define different inverse CDFs.

   (b) Sampling a binary outcome with probabilities produced from a logistic function taking $V_{Y|\operatorname{do}(T)}$ as input and conditioning on the given univariate $T$. The coefficient on $T$ can be chosen to impose the desired odds ratio.

   (c) Sampling from the univariate NSF learnt during the FF training, but with a user-defined location translation parameter representing the ATE and conditioning on the given treatment $T$. This method exploits the flexible margin distribution learnt for $T = 0$ during FF training, but allows to choose a different ATE for the location-translation effect produced by $T = 1$.

## D   Experimental Details

All experiments were run on a MacBook (16-inch, 2021) with an M1 Max chip and 32GB memory using the CPU.

### D.1   Simulated Data Experiments

The simulated data generated for the inference experiments was generated using the `causl` package written in R, which was called in Python via the `rpy2` package (Evans, 2021; Evans and Didelez, 2024).

The covariates were either selected to be binary (marginally distributed according to Bernoulli($p = 0.5$) or continuous (marginally distributed according to Gamma($\mu = 1, \phi = 1$). The marginal causal effect was chosen to be a linear Gaussian; $Y \mid \operatorname{do}(T) \sim \mathcal{N}(T, 1)$.

The underlying data generating process uses a multivariate Gaussian copula to generate dependencies between the marginal covariates and the causal effect. The Spearman correlation matrix used to generate the data for models $M_1$ and $M_2$ is

$$\mathbf{R}_4 = \begin{pmatrix} 1.0 & 0.5 & 0.3 & 0.1 & 0.8 \\ 0.5 & 1.0 & 0.4 & 0.1 & 0.8 \\ 0.3 & 0.4 & 1.0 & 0.1 & 0.8 \\ 0.1 & 0.1 & 0.1 & 1.0 & 0.8 \\ 0.8 & 0.8 & 0.8 & 0.8 & 1.0 \end{pmatrix}$$

and the correlation matrix used to generate data for model $M_3$ is

$$\mathbf{R}_{10} = \begin{pmatrix} 1.0 & 0.3 & 0.4 & 0.5 & 0.1 & 0.2 & 0.7 & 0.5 & 0.4 & 0.5 & 0.5 \\ 0.3 & 1.0 & 0.3 & 0.6 & 0.3 & 0.4 & 0.4 & 0.6 & 0.3 & 0.2 & 0.5 \\ 0.4 & 0.3 & 1.0 & 0.5 & 0.2 & 0.1 & 0.1 & 0.0 & 0.4 & 0.4 & 0.5 \\ 0.5 & 0.6 & 0.5 & 1.0 & 0.2 & 0.2 & 0.5 & 0.5 & 0.3 & 0.4 & 0.5 \\ 0.1 & 0.3 & 0.2 & 0.2 & 1.0 & 0.1 & 0.5 & 0.6 & 0.2 & 0.4 & 0.5 \\ 0.2 & 0.4 & 0.1 & 0.2 & 0.1 & 1.0 & 0.0 & 0.4 & 0.2 & 0.5 & 0.5 \\ 0.7 & 0.4 & 0.1 & 0.5 & 0.5 & 0.0 & 1.0 & 0.4 & 0.4 & 0.4 & 0.5 \\ 0.5 & 0.6 & 0.0 & 0.5 & 0.6 & 0.4 & 0.4 & 1.0 & 0.4 & 0.4 & 0.5 \\ 0.4 & 0.3 & 0.4 & 0.3 & 0.2 & 0.2 & 0.4 & 0.4 & 1.0 & 0.4 & 0.5 \\ 0.5 & 0.2 & 0.4 & 0.4 & 0.2 & 0.5 & 0.4 & 0.4 & 0.4 & 1.0 & 0.5 \\ 0.5 & 0.5 & 0.5 & 0.5 & 0.5 & 0.5 & 0.5 & 0.5 & 0.5 & 0.5 & 1.0 \end{pmatrix},$$

where the later rows/columns are indexed by the causal effect ranks, $V_{Y|\,\mathrm{do}(T)}$, and the earlier rows/columns correspond to the Spearman correlation matrix between the ranks of the covariates, $V_{\mathbf{Z}}$.

The propensity model for $M_1$ and $M_2$ was a sigmoid

$$p_{T|\mathbf{Z}}(t \mid \mathbf{z}) = \mathrm{Sigmoid}(-0.3 + 0.1z_1 + 0.2z_2 + 0.5z_1 z_2 - 0.2z_3 + z_4),$$

and $M_3$ was parameterised by

$$p_{T|\mathbf{Z}}(t \mid \mathbf{z}) = \mathrm{Sigmoid}(-0.3 + 0.1z_1 + 0.2z_2 + 0.5z_3 - 0.2z_4 + z_5$$
$$+ 0.3z_6 - 0.4z_7 + 0.7z_8 - 0.1z_9 + 0.9z_{10}).$$

### D.1.1 Hyperparameters and Runtime

The hyperparameters and runtime for the simulated inference datasets are presented in Table 4. In these cases, the models were trained with a default set of hyperparameters.

Table 2: Runtime and hyperparameters for fitting 25 different runs of each model, with a datasize of 15,000.

| Model | Total Runtime | RQS Knots | Flow Layers | Learning Rate | NN Width | NN Depth |
|-------|---------------|-----------|-------------|---------------|----------|----------|
| $M_1$ | 45.4 mins | 8 | 5 | 5e-3 | 50 | 4 |
| $M_2$ | 44.1 mins | 8 | 5 | 5e-3 | 50 | 4 |
| $M_3$ | 66.3 mins | 8 | 5 | 5e-3 | 50 | 4 |

## D.2 Additional Results

### D.2.1 Logistic Benchmark Simulation

To demonstrate the FF ability to generate discrete outcomes from known marginal logistic models, we ran the following experiment. First, data of the following form

$$Z \sim \mathcal{N}(\mu = 0, \sigma = 2)$$
$$V_Z, V_{Y|\,\mathrm{do}(T)} \sim c_{\mathrm{Gaussian}}(\rho = 0.8)$$
$$Y \mid \mathrm{do}(T) \sim \mathcal{N}(\mu = 2X, \sigma = 1)$$

was generated from the `causl` package. It was then fitted with a FF using the same hyperparameters as in Appendix D.1.1. We then generated samples from a custom logistic CDF such that

$$Y \mid \mathrm{do}(T) \sim \mathrm{Bernoulli}(p = \mathrm{Sigmoid}(2X - 1)).$$

A dataset size of $N = 1000$ was generated from the model, and fit using two models. The first is a Bernoulli outcome regression model, and the second uses inverse propensity weighting (IPW) to estimate the logistic parameters instead. The outcome regression (OR) estimates are biased indicating a clear confounding effect, whereas the IPW estimates comfortably contain the true parameters within their $2\sigma$ bounds. These results are presented in Table 3.

Table 3: Mean and 2-sigma confidence interval of the logistic parameter estimates. The "ground-truth" estimates are contrasted alongside the IPW estimates and OR methods, the latter of which demonstrates clear data confounding.

| Model | Parameter 1 | Parameter 2 |
|---|---|---|
| Ground Truth | $-1$ | $+2$ |
| Robust IPW | $\mathbf{-0.88 \pm 0.38}$ | $\mathbf{1.6 \pm 0.48}$ |
| Outcome Regression | $-1.59 \pm 0.24$ | $3.16 \pm 0.34$ |

$$
\begin{bmatrix} V_{\boldsymbol{W}} \\ Y \mid \boldsymbol{W}, \mathrm{do}(T) \\ \boldsymbol{V}_{\overline{\boldsymbol{W}}} \end{bmatrix} \longrightarrow \begin{pmatrix} \mathbb{I}(\cdot) \\ \mathcal{F}^{-1}_{Y \mid \boldsymbol{W}, \mathrm{do}(T)}(\cdot) \\ \mathbb{I}(\cdot) \end{pmatrix} \longrightarrow \begin{bmatrix} V_{\boldsymbol{W}} \\ V_{Y \mid \boldsymbol{W}, \mathrm{do}(T)} \\ \boldsymbol{V}_{\overline{\boldsymbol{W}}} \end{bmatrix}
$$

$$
\begin{bmatrix} \boldsymbol{U}_{1:d} \\ U_{d+1} \\ \boldsymbol{U}_{d+2:(D+2)} \end{bmatrix} \longleftarrow \mathrm{NSF} \begin{pmatrix} c(\boldsymbol{v}_{\boldsymbol{W}}) \\ c(v_{Y \mid \boldsymbol{W}, \mathrm{do}(T)}) = 1 \\ c(\boldsymbol{v}_{\overline{\boldsymbol{W}}} \mid v_{Y \mid \boldsymbol{W}, \mathrm{do}(T)}, \boldsymbol{v}_{\boldsymbol{W}}) \end{pmatrix}
$$

Figure 6: Structure for learning a Frugal Flow with a heterogeneous treatment effect. This enforces that $\boldsymbol{v}_{\boldsymbol{W}} \perp\!\!\!\perp v_{Y \mid \boldsymbol{W}, \mathrm{do}(T)}$ and that the copula density of $\boldsymbol{v}_{\boldsymbol{W}}$ can be inferred via the factor $c(\boldsymbol{v}_{\boldsymbol{W}} \mid v_{Y \mid \boldsymbol{W}, \mathrm{do}(T)})$

### D.2.2 Heterogeneous Treatment Effects

In the main paper, we comment on the ability of the model to simulate data from distributions where the causal effect is a marginal quantity taken over the entire covariate set

$$
p_{Y \mid \mathrm{do}(T)}(y \mid t) = \int_{\mathcal{Z}} d\boldsymbol{z} \, p_{Y \mid \boldsymbol{Z}, \mathrm{do}(T)}(y \mid \boldsymbol{z}, t) \, p_{\boldsymbol{Z}}(\boldsymbol{z}). \tag{8}
$$

However, one may wish to simulate from more complex heterogeneous treatment effect models. Consider a stationary treatment with pretreatment covariate set $\boldsymbol{Z} = \{\boldsymbol{W}, \overline{\boldsymbol{W}}\}$ where $\boldsymbol{W} \subset \boldsymbol{Z}$ and $|\boldsymbol{Z}| = D$, $|\boldsymbol{W}| = d$, and $|\overline{\boldsymbol{W}}| = D - d$. We proceed considering the case where $0 < d < D$.

Interest may lie in the causal treatment margin **conditional** on the subset of variables $\boldsymbol{W}$:

$$
p_{Y \mid \boldsymbol{W}, \mathrm{do}(T)}(y \mid \boldsymbol{w}, t) = \int_{\overline{\mathcal{W}}} d\overline{\boldsymbol{w}} \, p_{Y \mid \boldsymbol{Z}, \mathrm{do}(T)}(y \mid \boldsymbol{w}, \overline{\boldsymbol{w}}, t) \, p_{\overline{\boldsymbol{W}} \mid \boldsymbol{W}}(\overline{\boldsymbol{w}} \mid \boldsymbol{w}) \tag{9}
$$

which we call the conditional treatment margin. We infer this effect by constructing a Frugal Flow which ensures that the pretreatment covariate joint $p_{\boldsymbol{Z}}(\cdot)$ is correctly inferred and that the conditional treatment margin ranks are uniformly distributed. A modified version of the Frugal Flow illustrated Figure 2 is used to account for this change. The choice of $\mathcal{F}^{-1}_{Y \mid \boldsymbol{W}, \mathrm{do}(T)}(\cdot)$ can be left to the user for inferring the Frugal Flow. For simulating benchmarks, the conditional treatment margin can be free set to any valid CDF, for example:

$$
Y \mid \boldsymbol{W}, \mathrm{do}(T) \sim \mathcal{N}(\mu = g(\boldsymbol{W}, T), 1)
$$

where $g(\cdot) : \mathbb{R}^d \times \mathcal{T} \to \mathbb{R}$ can be chosen to encode arbitrary heterogeneity in treatment effects.

### D.3 Real-World Data Benchmarks

### D.3.1 Lalonde Temporary Employment Program

The Jobs dataset by LaLonde is a benchmark in causal inference studies, where job training serves as the treatment and the outcomes are post-training income and employment status. Originating from the National Support Work Demonstration (NSW), this randomized controlled trial (RCT) examines the impact of a temporary employment program (i.e. the treatment) in the US on participants' income levels (LaLonde, 1986). Due to its design the treatment assignment is random, eliminating unobserved confounding. The measured features, all recorded in 1975, are:

- an individual's age in years;
- the number of years an individual spent in education;
- whether an individual is black;
- whether an individual is hispanic;
- an individual's marital status (1 if married, 0 otherwise);
- whether an individual has a high school degree.

The outcome is the individual's real earnings in 1978.

### D.3.2   401(k) Eligibility

The 401(k) savings plans dataset has been analysed in a variety of studies. We use it to investigate the impact of eligibility to enroll on the increase in net assets.

The dataset includes 9,915 individuals with the following variables measured:

- age of the individual in years;
- income of the individual;
- years of education the individual has completed;
- size of the individual's family;
- indicator variable of whether the individual is married (1 for married, 0 otherwise);
- indicator of whether there are two earners in the household (1 if two earners, 0 otherwise);
- membership of a defined benefit pension scheme (1 if true, 0 otherwise);
- eligibility for Individual Retirement Allowance (IRA) (1 if true, 0 otherwise);
- homeownership status of the individual (1 if true, 0 otherwise).

### D.3.3   Causal methods used for benchmarking FF inference

Section 4.1 reports FF ATE inference performance in a simulated setting comparing to a number of methods.

- Outcome regression (OR) ATE results are obtained by regressing $Y$ on the treatment $T$ together with the covariates $Z$ in the different scenarios and reporting the coefficient and confidence interval on $T$.
- Propensity score matching is implemented using the R package `MatchIt` for estimating the ATE (Stuart et al., 2011).
- Causal normalising flow (CNF) (Javaloy et al., 2024) is trained using the causal abductive model with one layer, that the paper reports to be the best-performing model variation (Paragraph 6.1). We use the hyperparameter settings recommended in the package and do not perform hyperparameter tuning. For the flow architecture, we use neural spline flows, as the paper reports in Appendix D.3 that they yield a better performance than simple masked autoregressive flows, plus this resembles our architecture choice in Frugal Flows. We do not add uniform independent noise to the binary inputs as recommended in paragraph 3.1 as we find this worsens the ATE estimates for the model.

### D.3.4   Causal methods used for validating FF as a benchmark

Similar to Parikh et al. (2022) we use a variety of different causal inference methods to validate the generated benchmark samples of our model.

- Propensity score matching is implemented using the R package `MatchIt` for estimating the ATE (Stuart et al., 2011).
- Causal BART is implemented via the R package `dbarts` using default hyperparameters (Dorie et al., 2024).

- The double machine learning methods are implemeneted using the Python package `EconML` (Research, 2019) using the scikit-learn's machine learning API for the same. For GBT DML, we used the method with 100 trees, and the linear DML used ridge regression (Pedregosa et al., 2011).
- `EconML` was also used for implementing the S-, T- and X-learners also using scikit-learn's ML API to for gradient boosting trees and ridge regression.
- TMLE was implemented using the `zepid` Python package (Zivich, 2020).

### D.3.5 Hyperparameters and Runtime

For both datasets, a random hyperparameter search was conducted by choosing the hyperparameter set which minimised the validation loss, given a train/test data split of $9/1$. The total number of neural network of the hyperparameter tuned Frugal Flow for the Lalonde and e401(k) datasets are 485243 and 106969 respectively.

Table 4: Runtime and hyperparameters for fitting both a Frugal and Propensity Flow to the Lalonde and e401(k) data.

| Benchmark | Runtime | Knots | Flow Layers | Learning Rate | NN Width | NN Depth |
|---|---|---|---|---|---|---|
| Lalonde | 1.2 mins | 4 | 9 | 6.3e-3 | 50 | 10 |
| e401(k) | 4.9 mins | 5 | 2 | 2.6e-3 | 34 | 3 |

### D.3.6 Realism of Datasets

We conducted additional validation of the proposed Frugal Flows method to enhance its robustness in comparison to current state-of-the-art methods such as Credence (Parikh et al., 2022) and RealCause. Credence allows for the exact specification of conditional average treatment effect (CATE) in generative samples, whereas RealCause adjusts the causal effect *post hoc*, preventing it from realistically modelling a null hypothesis where the average treatment effect (ATE) is zero. Additionally, Frugal Flows and Credence both model unobserved confounding, a feature absent in RealCause, making Credence the more suitable method for direct comparison.

We ran the benchmarking simulations in Section 4.2 with Credence, using its default parameters, evaluating model performance on Lalonde and the e401(k) dataset. We compared the correlation matrices of the pretreatment covariates and outcomes for the original data, Frugal Flows-generated data, and two Credence-generated datasets with different causal constraints. The results, illustrated in Figure 7 and Figure 8, show that Frugal Flows produce samples closely resembling the original data, especially for the larger e401(k) dataset.

While Credence also performs well on the e401(k) dataset, altering its causal constraints significantly affects the covariate dependencies. In contrast, Frugal Flows optimise the model once, allowing for causal constraint modifications without altering the covariate joint distribution or propensity score.

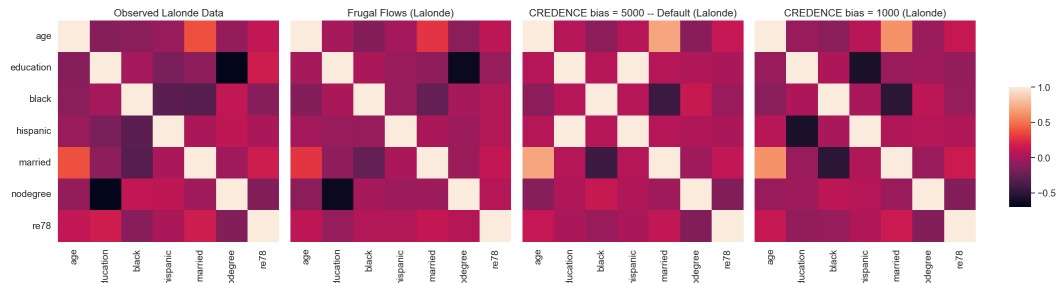

Figure 7: **Lalonde**: Correlation matrices across covariates and the outcome (`re78`), comparing the second moments of distributions for the Lalonde observed real data, as well as synthetic data generated by a trained Frugal Flow (2nd column) and Credence (3rd column) models. The comparison is further extended to models with default settings and those with modified bias rigidity (4th column).

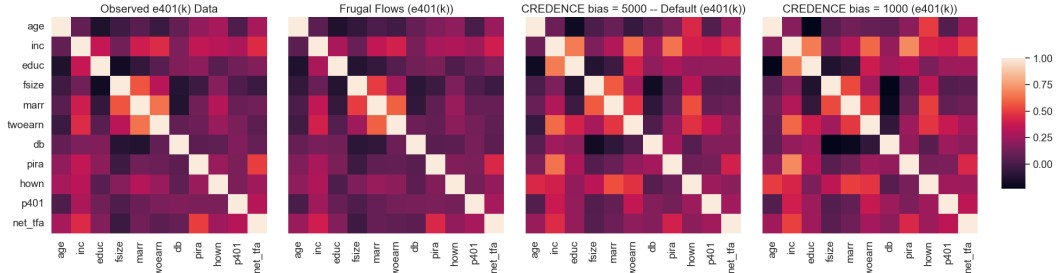

Figure 8: **e401(k)**: Correlation matrices across covariates and the outcome (`net_tfa`), comparing the second moments of distributions for the e401(k) observed real data and synthetic data generated by trained Frugal Flow (2nd column) and Credence (3rd column) models. The comparison is further extended to models with default settings and those with modified bias rigidity (4th column).

### D.3.7 Loss Optimisation

In training, we perform a train-val split and use a "patience" criterion on the validation loss as a criterion to stop the training. Namely, we monitor the validation loss and stop training if the validation loss does not improve for a specified number of epochs (we set the patience value to 100). This aims to prevent overfitting and saves computational resources by not continuing training unnecessarily. It is standard in machine learning model training and was implemented in the FlowJax package that we use as code-base to build the Frugal Flow package from.

The observational likelihood losses during model training for both real-world datasets are presented in Figures 9 and 10.

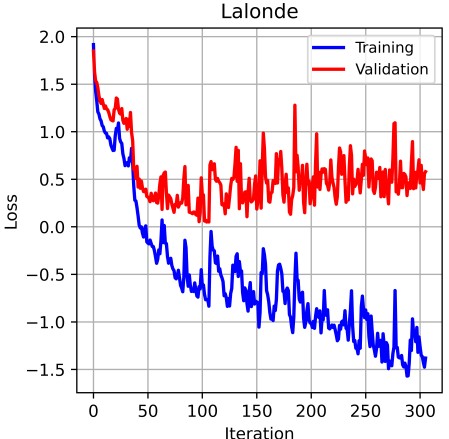

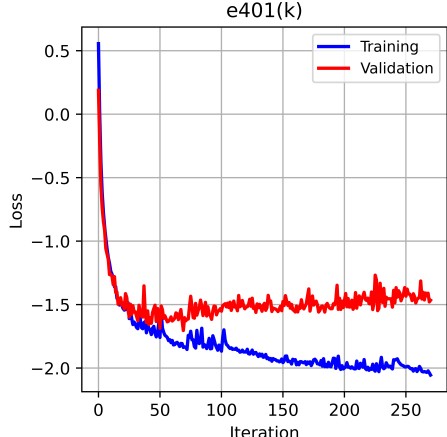

Figure 9: Training and validation losses when fitting a Frugal Flow to the **Lalonde** dataset using optimal hyperparameters and a "patience" setting of 200 iterations for illustrative purposes.

Figure 10: Training and validation losses when fitting a Frugal Flow to the **e401(k)** dataset using optimal hyperparameters and a "patience" setting of 200 iterations for illustrative purposes.

