# OpenReview forum: "Marginal Causal Flows for Validation and Inference"
_NeurIPS.cc/2024/Conference — NeurIPS 2024 poster_

### Official Review · Reviewer_J42v · 2024-07-03

**Soundness:** 3
**Presentation:** 3
**Contribution:** 3
**Rating:** 7
**Confidence:** 3

**Summary:**

This paper introduces _Frugal Flows_ a method that learns the data distribution of data for causal effect estimation; namely outcome $Y$, binary treatment $X$ and pretreatment covariates $\mathbf{Z}$.
Through a combination of frugal parametrisation, normalizing flows and copulas, separate components for the marginal causal effect $p_{Y| do(X)}$, the probability integral transforms of $\mathbf{Z}$ and the propensity score are leaned.
(The components of) the learned model, can be used for (i) estimating the marginal effect and (ii) to generate synthetic data with a fixed marginal effect for benchmarking other causal inference methods.
In the second application the component for the marginal effect is switched out for another with desired properties.
(i) is demonstrated on small synthetic datasets.
(ii) is demonstrated by fitting FFs to two real-world datasets and generating synthetic data with adjusted properties.

**Strengths:**

- The paper is well-written.
- It tackles an important problem in causality research. Since, randomized data is hard and expensive to get, many causal methods are only evaluated on synthetic data and generating realistic/semi-synthetic data is hard. This paper makes a great contribution towards improving synthetic data generation. If the code for the method is provided in a user-friendly manner, I could see this having a big impact on the causality community.

**Weaknesses:**

- Normalizing Flows have been used in the causal modelling context before (see [1, 2]). While prior works solve different problems (the inferred latents correspond to exogenous variables of an SCM, not directly applied to causal effect estimation), I think it would still be valuable to contrast this work to what has been done before for future reference in the literature.
- L59: The basic causal assumptions aren't explicitly stated. What are the causal assumptions on $X$, $Y$ and $\mathbf{Z}$? It seems like the method wouldn't hold if $\mathbf{Z}$ was a mediator (I suppose the equation after L60 wouldn't hold). A reference to a 500+ page book is given for the assumptions, which feels like a slap in the face for the reader.
- The notation for interventional distributions is confusing: what's the difference between using an asterisk and explicitly using the do-notation? In the equation after L60, the LHS seems to be an interventional quantitiy (asterisk, but no do-notation), whereas Equation (1) has the do-notation, but no asterisk. Do the two notation elements mean different things?
- I think this paper would greatly benefit from a visual abstract showing how the different flows and distributions come together. Maybe this is something that could be added for the camera-ready.


Minor:

- L201: typo

[1] Javaloy et al. "Causal normalizing flows: from theory to practice." NeurIPS 2023

[2] Wendong et al. "Causal component analysis" NeurIPS 2023

**Questions:**

- Fig. 1: What's the meaning of the red undirected edges. Does it mean they could go either way and/or they could be confounded? Please specify this somewhere in the paper or appepndix.
- Def. 1, App. A: I struggle to understand this definition. The cartesian product on the RHS makes sense to me, but what does the LHS of the equation mean? What is the "x" operation between functions? Do the two functions map to the same space?
- Fig. 2: Why is the first line needed? If I understand correctly, this just makes all pretreatment variables uniform. Couldn't you put $\mathbf{Z}$ directly in the second line?
- You show synthetic data generation based on two datasets in Sec. 4.2. Why couldn't you use the same datasets to test causal effect estimation in Sec. 4.1?
- How many datapoints are in the Lalonde dataset?
- In training, how did you check whether the training has succeeded? I suppose you minimize the log-likelihood, how did you define "good enough"? I'm asking because the training seems pretty fast (App. D2.4). What's the total number of parameters for each of the datasets?

**Limitations:**

- The biggest limitation of the work seems to be the requirements for the dataset size, needed to train normalizing flows. The authors mention this in Sec. 4.1. There could be some applications that have enough data for using Frugal Flows for causal effect estimation (e.g. online businesses with many customers, or well-curated medical datasets like the ones from healthcare providers in Isreal). However, for most applications the data won't be enough.

---

> ### Author Rebuttal · Authors · 2024-08-07
>
> We thank the reviewer for their comprehensive assessment, thoughtful commentary, and suggestions. We share your aspiration that frugal flows can be engineered and presented in a user-friendly manner for the causality community.
> # Weaknesses
> We would like to address the comments under the Weaknesses section, responding in the order presented:
> 1) We appreciate the references shared by the reviewer. The Causal Normalizing Flows (CNFs) paper, also referenced by another reviewer, provides an interesting contrast to our method. While these papers tackle slightly different problems in inferring marginal effects, we are happy to include and discuss them in our literature review. Additionally, we implemented a variation of CNFs using default hyperparameters to estimate ATEs within our synthetic experiments in section 4.1. These are presented in Figure 1 of the attached document to the rebuttal. CNFs showed higher variance in ATE estimates than frugal flows, which is expected as CNFs are not designed for causal effect estimation. We also provided figures showing the observational likelihood loss for real-world benchmarks and synthetic experiments
> 2) A key assumption for our model is that the covariate set $\mathbf{Z}$ must be pretreatment covariates. You are correct that the parameterization would not hold if a $Z_{d} \in \mathbf{Z}$ were a mediator. We will explicitly summarize the required assumptions for our method in our paper.
> 3) Thank you for the suggestion regarding the labeling of interventional distributions! The asterisk notation, taken from Evans and Didelez (2024), is used to contrast a wider range of distributions. However, in this paper, we consider a single static treatment model and are happy to use a simpler notation, specifically $do(X)$ operators, in an updated submission.
> 4) We appreciate the suggestion of a visual abstract and believe it would significantly improve the presentation and readability of our method. We are delighted to incorporate this addition into our paper.
>
> We also thank the reviewer for flagging up a typo, which we have corrected!
> # Questions
> In response to the reviewer’s specific questions:
>
> * Fig 1: There is an error in Figure 1. There should be directed edges pointing from each $Z$ to $Y$ ($Z \to Y$) and undirected edges between each of the $\mathbf{Z}$ variables. The undirected edges indicate that the edges could point either way. We will clarify this in the paper.
> * Variation independence (VI) is a highly desirable property for a parameterization, since it allows different components to be specified entirely separately.  This is extremely useful if one is trying to use a link function in a GLM, or to specify independent priors for a Bayesian analysis.  In addition, VI is important in semi-parametric statistics.  The definition simply states that the Cartesian product of the images is the same as the image of the joint map.  For example, in a bivariate gamma-distribution with positive responses, then $\mu_1 \in \mathbb{R}^+$ and $\mu_2 \in \mathbb{R}^+$ is a variation independent parameterization, since
> $$
> (\mu_1 \times \mu_2)(\Theta) = \mathbb{R}^+ \times \mathbb{R}^+ = \mu_1(\Theta) \times \mu_2(\Theta).
> $$
> However, if we replace $\mu_2$ with $\mu_2’ = \mu_2-\mu_1$ (for example), then although the range of this parameter is $\mathbb{R}$,
> $$
> (\mu_1 \times \mu’_2)(\Theta) = \{(x,y) : x > 0, y > -x\}  \neq \mathbb{R}^+ \times \mathbb{R} = \mu_1(\Theta) \times \mu’_2(\Theta).
> $$
> * Fig 2: We separate the training of the marginal densities of the pretreatment covariates to ensure that the normalizing flow on the second line represents a copula flow. Training in a single line would not enforce that the marginal densities of the copula flow are uniform (except $V_{Y|do(X)}$). Thus, the marginal flows and copula flow must be trained separately (see lines 175-178 in original submission). Training the multivariate flow on the empirical CDF of the covariates also provides flexibility to generate realistic synthetic data from populations with marginal densities differing from the training data while maintaining the underlying dependencies of the original dataset.
> * In their current form, we do not believe frugal flows can reliably infer ATEs from small to medium-sized datasets. Figure 3 in the attached material illustrates the sensitivity of the ATE estimate as a function of data size, using the synthetic data models $M_{1}$ and $M_{2}$ in section 4.1 (original paper). Our experiments show that ATE estimates are reliably inferred only at data sizes greater than $N \approx 10,000$. For this reason, we did not present ATE estimates for the real-world data, as they are both smaller than $10,000$. The e401(k) dataset is of similar size but with more than double the number of dimensions. This however does not impede us from generating realistic synthetic benchmarks as validated by Figure 2, Figure 3, and Table 2 in the document attached to the rebuttal. These provide empirical evidence that generated data is statistically similar to the original dataset.
> * The processed Lalonde dataset is of size $N = 614, D = 6$. We will add this information to the paper.
> * The parameter number of the frugal flow trained on the Lalonde and e401(k) data are 485243 and 106969 respectively. We will add this information to the paper.
> * In training, we perform a train-val split and use a “patience” criterion on the validation loss as a criterion to stop the training. Namely, we monitor the validation loss and stop training if the validation loss does not improve for a specified number of epochs (Chapter 5.5.2 in “Pattern recognition and machine learning” (Bishop and Nasrabadi, 2006)). We set the patience value to 100. This aims to prevent overfitting and saves computational resources by not continuing training unnecessarily. It is standard in machine learning model training and was implemented in the FlowJax package that we use as code-base to build the Frugal Flow package from.

---

> > ### Comment · Reviewer_J42v · 2024-08-13
> >
> > Thank you for providing a detailed response to my questions. I will keep my score of acceptance.

---

### Official Review · Reviewer_Cwuj · 2024-07-11

**Soundness:** 3
**Presentation:** 4
**Contribution:** 3
**Rating:** 7
**Confidence:** 4

**Summary:**

The paper introduces a generative modeling approach called Frugal Flows, designed to learn the data generation process with an explicit parametrization for the marginal causal effect of treatment on outcomes. Inspired by the frugal parametrization of marginal structural models, this approach models the marginal intervention distribution $p(Y|do(X))$ directly, rather than the joint distribution $p(Y|Z, do(X))$. This helps in preserving any constraints on the average treatment effect while flexibly modeling the data generation process. Frugal Flows employs copula flows to parameterize the model, accommodating constraints on the average causal effect and handling unobserved confounding during data generation. The authors validate the proposed method through experiments on both synthetic and real-world datasets, demonstrating its ability to generate realistic datasets with user-specified constraints.

**Strengths:**

* The paper's approach to validating causal models using simulated datasets is indeed impactful and relevant. It addresses a significant gap by allowing for general constraints on quantities of interest, such as average causal effect and unobserved confounding, during data generation. This capability is crucial because many prior generative modeling approaches for causal datasets either do not offer such flexibility or cannot ensure the preservation of these constraints, thus making this work a notable advancement in the field.

* The paper is well-written, with clear explanations in the background sections on frugal parametrization and flows, which help the reader grasp the proposed approach. The details of the approach are well-articulated, and the experimental results are presented effectively.

* The proposed approach is indeed novel. While it builds on established concepts like frugal parametrization, the specific application of normalizing flows for parametrization and its focus on average causal effect estimation represent a significant and innovative contribution.

**Weaknesses:**

My main concern with the work is the limited empirical validation of the proposed approach. Given that the primary contribution is the learning methodology rather than theoretical analysis, I would expect a more extensive set of experiments to validate its effectiveness. For example, prior research on generative modeling for causal inference, such as the work by [1], includes comprehensive experiments with various statistical tests to assess the realism of generated samples and a broader benchmarking of causal estimators. This paper would benefit from similar depth in its empirical evaluation.

It would nice if the authors can conduct similar experiments to asses whether learned generative model generates realistic samples and evaluate it on more datasets. Also, the authors should compare with the prior works [1, 2] as baselines to establish which approach is the best at capturing the underlying data generation process, and empirically validate their claim (Section 2.6) that the proposed approach would be better than prior works at capturing used-specified constraints on the average causal effect.

References

[1] Neal, Brady, Chin-Wei Huang, and Sunand Raghupathi. "Realcause: Realistic causal inference benchmarking." arXiv preprint arXiv:2011.15007 (2020).

[2] Harsh Parikh, Carlos Varjao, Louise Xu, and Eric Tchetgen Tchetgen. Validating causal inference methods. In International conference on machine learning, pages 17346–17358. PMLR, 2022.

**Questions:**

* A suggestion for the notation is that the authors could use $T$ instead of $X$ to denote the treatment variables in the paper. This way the notation would be less confusing, as $X$ represents a general random variable in Section 2.5 and Section 2.6

* Maybe there is a typo in Figure 2? The top row should be $\mathcal{F_{Z_i}}^{-1}(.)$   as we transforming the covariate $\{ Z_i \}$ to the correlated uniform variables $\{ V_{i} \}$

* I don't understand the Section 3.1.1 on copula flow for $X$ on $Z$. Why don't we directly model $p(X|Z)$ using a normalizing flow and why do we need to parametrize using copula flow as $p(X|Z)= p(X).c(X|Z)$?

**Limitations:**

Yes, the authors have adequately addressed the limitations of their work.

---

> ### Author Rebuttal · Authors · 2024-08-07
>
> We thank the reviewer for their detailed analysis of our paper. Before addressing the specific questions, we would like to comment on the important points raised in the weaknesses section.
>
> ## Weaknesses
> We agree that a more comprehensive validation of our proposed approach would strengthen the narrative and enhance the perception of frugal flows as a reliable method for improving synthetic data generation. The two papers referenced by the reviewer, which we also mention in our paper, provide algorithms most comparable to our proposal. Credence (Parikh et al, 2023) allows exact specification of the CATE in generative samples. In contrast, RealCause (Neal et al, 2021) only allows users to scale the causal effect post-hoc, which hinders its ability to model a null hypothesis of ATE=0, where all outcomes are scaled to zero. Additionally, frugal flows’ ability to model unobserved confounding is shared with Credence but not with RealCause. Therefore, we chose Credence as the more appropriate method for comparison.
>
> Following the reviewer’s suggestions, we reran the benchmarking simulations in section 4.2 with Credence, using the default parameters specified in their package. We present an illustration of the correlation matrix of the pretreatment covariates and the outcome for the original data, data samples from frugal flows, and two distinct data samples from Credence trained with different causal constraints. We refer the reviewer to Figure 1 and Figure 2 in the document attached to the rebuttal. Frugal flows generate samples closely resembling the original data, particularly for the larger e401k dataset.
>
> The samples generated by Credence also perform well in generating realistic samples for the e401k dataset. However, we note that altering the causal effect rigidity significantly impacts the covariate joint and introduces dependencies not present in the original data. A key advantage of frugal flows is the need to optimize the model fitting only once, allowing for direct modification of causal constraints without affecting the covariate joint and propensity score.
>
> We also conducted the same multivariate statistical tests used by Neal et al. (2021) to validate RealCause. These tests include the Maximum Mean Discrepancy test (Gretton et al., 2012), the energy test (Szekely & Rizzo, 2013), the Friedman-Rafsky test (Friedman & Rafsky, 1979), and the k-nearest neighbor (kNN) test (Friedman & Rafsky, 1983), all implemented in the $\texttt{torch-two-sample}$ Python library. Across both datasets, the tests suggest that frugal flows generate samples that are more comparable to the original data than those generated by Credence. In most tests, there is not sufficient evidence to claim that samples from frugal flows are distinguishable from the original data.
>
> Regarding benchmarks, we conducted experiments on two real-world datasets (Lalonde, e401k), which is the same number as the Credence paper (Lalonde, Project STAR). Due to time constraints, we could not obtain results by the end of the rebuttal period but are happy to run additional experiments to validate and contrast frugal flows against Credence using the Project STAR dataset, which was also explored in the Credence paper. This would put us on par with the RealCause paper, which validates their method using three real world datasets.
>
> ## Questions
> In response to the reviewer’s questions, we provide the following answers:
>
> 1) We fully agree with your suggestion to change X to T when referring to treatments in the paper. Although we have not made this change in the current rebuttal to avoid confusion among other reviewers, we have incorporated this change in the manuscript.
> 2) Thank you for spotting the typo! You are absolutely correct, and we have corrected the error in the manuscript.
> 3) You’re right here—one could directly model $p_{X|\mathbf{Z}}$ using a normalizing flow, which would be a valid frugal model. We model $c_{X|\mathbf{Z}}$ to encode a degree of unobserved confounding in the generated data by sampling the ranks $U_{X|\mathbf{Z}}$ and $U_{Y|\text{do}(X)}$ from a non-independence copula. Assuming ignorability, these ranks would be independent. However, unobserved confounders imply marginal dependency between these ranks. Sampling from a copula replicates this effect, as demonstrated in the far-right plots in Figures 3 and 4 in the original submission. We discuss this in lines 260-265, explaining how to simulate unobserved confounding.

---

> > ### Comment · Reviewer_Cwuj · 2024-08-11
> >
> > Thanks for the detailed response! My questions have been addressed and I have increased my rating accordingly!

---

### Official Review · Reviewer_SZi8 · 2024-07-12

**Soundness:** 3
**Presentation:** 3
**Contribution:** 3
**Rating:** 6
**Confidence:** 3

**Summary:**

This paper proposed a generative model called Frugal Flows making use of copula flows to infer about marginal causal effects by simulating the data generating process.

**Strengths:**

- The problem of inferencing marginal causal effects is an interesting and important problem
- The idea of using generative models to estimate the marginal effects in the paper is interesting

**Weaknesses:**

See questions.

**Questions:**

I have two question --

1. A highly related (and I suspect might be viewed as a "dual" appproach to your Frugal Flow) is the statistical matching (e.g., bipartite matching) to estimate the average treatment effect. It would be very informative to compare this as one of the baselines in your benchmarking and validation, as this also shed lights on how these two different schools of causal inference may (or may not) converge on ATE estimation.

2. I think it is good to use real data for benchmarking/validation (but perhaps benchmarking is a bit strong here since only two datasets were used), but usually it is unclear how to interpret the results since the ground truth is unknown. Can you design and run some controlled synthetic experiment to verify the model?

**Limitations:**

See questions.

---

> ### Author Rebuttal · Authors · 2024-08-07
>
> We appreciate the reviewer’s comments and suggestions for our work. Indeed, we agree that frugal flows are an interesting addition to inference algorithms for estimating marginal causal densities in large datasets using Normalizing Flow models.
>
> In addition, we believe a key contribution of frugal flows lies in their ability to enable users to learn representations of realistic datasets and precisely customize causal features such as the ATE, degree of unobserved confounding, propensity score, and simulating from discrete outcomes. When referring to model benchmarking, we propose that frugal flows can generate realistic datasets with customizable causal features for benchmarking purposes; being able to simulate realistic data with a ground truth is clearly much more useful for benchmarking novel causal methods, than using real datasets with unknown underlying causal effects (see Section 2.6 in the original submission).
>
> In response to the reviewers questions, we sincerely appreciate your suggestions for improving the scope of our inference experiments. We have made a couple of modifications and additional experiments based on your suggestions:
> 1) We agree that Section 4.1 would be more compelling if frugal flows were contrasted with other causal inference methods, rather than linear regression, which was initially included to demonstrate the complexity of the confounding. Consequently, we have added results from a statistical matching algorithm as suggested. Additionally, we present ATE estimates calculated using an alternative method, **Causal Normalizing Flows** (Javaloy et al, 2023), recommended by other reviewers. The matching algorithms yield results consistent with frugal flows, whereas the CNFs produce ATE estimates with higher variance, and in some cases a bias, compared to both statistical matching and frugal flows.
> 2) In response to the desire for more comprehensive synthetic experiments, we demonstrate that with sufficiently large data sizes, they can accurately identify the true ATE across a range of simulated data. These results are presented in Table 1 (in the document attached to the rebuttal) for models with an ATE of 1. We have also rerun similar experiments with a true underlying ATE of 5, incorporating each of the algorithms suggested by the reviewers.

---

> > ### Comment · Reviewer_SZi8 · 2024-08-13
> >
> > Thank you for addressing my concerns.

---

### Official Review · Reviewer_PATR · 2024-07-12

**Soundness:** 3
**Presentation:** 2
**Contribution:** 3
**Rating:** 7
**Confidence:** 3

**Summary:**

This work proposes to leverage existing neural density estimators (specifically, normalizing flows) to exploit a newly-proposed "frugal parametrization" that can capture the causal marginal distribution of an underlying causal model. Under this parametrization, the authors show how to specify and train each component of the model, and thus train the proposed Frugal-Flows to match the observational distribution as closely as possible, while being able to tune the marginal causal effect present in the generated data.

This way, frugal flows can be used to generate synthetic causal benchmarks that closely represent the _observational_ data while having more difficult-to-estimate causal effects, putting existing approaches to the test.

**Strengths:**

- **S1.** The proposed frugal flows provide a way of generating new datasets that can be challenging from a causal-inference point of view, which I believe _important_ to test new and existing methods.
- **S2.** The construction of the proposed architecture is quite rich in details.
- **S3.** I find the frugal parametrization conceptually quite interesting.
- **S4.** The authors motivate different scenarios for frugal flows in Sec. 3.2, as well as empirically show positive results on some synthetic and real-world scenarios.

**Weaknesses:**

- **W1.** I find the frugal parametrization to be extremely under-explained, relying too much on the reader having full knowledge of the referenced work. Similarly, there is little to no explanation/intuition on why the frugal parametrization would properly capture the marginal causal distributions.
- **W2.** The lack of explanations also applies to other concepts, e.g., "conditional ignorability" (line 39) "variation independence" (line 82, and I know the definition is later in App. A), or why copula-based flows would target conditional causal effects instead of marginal causal ones (line 182). (similar with lines 221 and 229)
- **W3.** There are no mention to related works that propose similar ways of constructing causal benchmarks. From a 1-min search in google scholar, I already found some likely relevant works: [Work 1](https://arxiv.org/abs/2406.08311), [Work 2](https://arxiv.org/abs/2011.15007).
- **W4.** I find the experiments a bit underwhelming, specially those from Section 4.1. The authors should at least show how is the fitting of the observational likelihood, and if they want to show the capabilities of frugal flows for causal inference (and not only causal-benchmark generation), they should compare with other methods like [Causal Normalizing Flows](https://arxiv.org/abs/2306.05415).

**Questions:**

- **Q1.** I am not sure that I understand what does the dotted red line represent in the boxplots.
- **Q2.** Doesn't the statement in lines 291-294 directly contradict what you say later in lines 296-297?
- **Q3.** Why is Figure 1 placed there?

**Limitations:**

I think limitations are properly discussed.

---

> ### Author Rebuttal · Authors · 2024-08-07
>
> We are grateful for the reviewer’s feedback and suggestions for our submission. Below are our responses to the noted weaknesses and questions:
> # Weakness 1
> We acknowledge that the frugal parameterization was briefly introduced. Due to page constraints, we provided a brief overview in the main text and a more detailed discussion in the appendix, drawing largely from Evans and Didilez (2024). In response to the reviewer’s comments, we will reorganize section 2.2 to better explain frugal models, the relevance of copulas, and their focus on the causal margin. We will also enhance the appendix with a comprehensive introduction for those interested in technical details.
> Regarding why the frugal parameterization captures the marginal causal distribution when the dependency measure is parameterised by a multivariate copula, consider the copula for the distribution of $\mathbf{Z}$ and $Y$ conditional on $X$:
> $$c(F_{Y|do(X)}, F_{Z_1|X},\dots, F_{Z_{D}|X}).$$
> For an intervened distribution, all pretreatment covariates $\mathbf{Z}$ are marginally independent of $X$, simplifying the copula to
> $$ c(F_{Y|do(X)}, F_{Z_1},\dots, F_{Z_{D}}),$$
> and so the intervened joint density becomes
> $$p(\mathbf{Z},~Y \mid do(X)) = p_{Y|do(X)} \cdot \prod_{d=1}^{D} p_{Z_d} \cdot c(F_{Y|do(X)}, F_{Z_1},\dots, F_{Z_{D}}),$$
> where $p_{Y|do(X)}$ is the marginal causal effect of $X$ on $Y$. The final density required to parameterise the observational distribution is the propensity score $p_{X|\mathbf{Z}}$, which does not affect the aforementioned marginal densities in the observational model (see Chapter 10 in "Information and Exponential Families"(Barndorff Nielsen, 1978)). We hope this clarifies why frugal parameterizations target marginal causal quantities and will add these clarifications to the manuscript.
> # Weakness 2
> We agree that improved clarity in definitions and terminology would enhance readability and are happy to make these additions to the manuscript. We use the definition of conditional ignorability (equivalent to conditional exchangeability) stated in “Causality: Models, Reasoning and Inference” (Pearl, 2009), where the marginal potential outcomes are independent of the treatment, conditional on the observed covariates.
>
> Regarding why copula-based flows target the conditional effect, copula flows require all marginal flows to be trained independently of the copula likelihood. This is because copula flows infer a joint density within a unit hypercube and do not constrain marginal densities to be uniformly distributed. Thus, empirically uniform data must be provided to generate samples from a copula. Since one of the marginal densities being trained is a conditional quantity ($\mathcal{F}_{Y|X}$), training this flow independently targets the conditional of $Y$ on $X$. Frugal flows target the marginal causal effect as they are trained jointly with the copula flow, with the causal effect density constrained to be uniformly distributed, as outlined in Figure 2 of the original submission.
> # Weakness 3
> Our aim was to contrast our contribution to generating synthetic causal benchmarks against other methods in the literature, as discussed in section 2.6. We reference three key papers on line 200, including the second work the reviewer mentioned. We appreciate the reviewer sharing Work 1, which we were not aware of and was submitted to arXiv after our paper submission! However, Work 1 focuses more on validating structural causal benchmarks, whereas Work 2 (and the other two papers we reference) focus on generating observational data with customizable causal effects.
>
> Credence allows exact specification of the CATE in generative samples. In contrast, RealCause only allows users to scale the causal effect post-hoc, which hinders its ability to model the null hypothesis of the ATE=0, because every outcome is scaled to zero. Additionally, frugal flows’ ability to model unobserved confounding is shared with Credence but not with RealCause. Therefore, we chose Credence as the more appropriate method for comparison.
>
> We assessed the realism of samples from frugal flows for the Lalonde and e401k datasets, comparing them to those from Credence. These results are presented in Figures 1 & 2 (in the attached material) showing the correlation matrices of covariates and outcomes for frugal flow and Credence samples against the original data. We find that Credence's fitting process is sensitive to the causal effect one wishes to constrain, requiring hyperparameter optimization for every setting, while frugal flows need optimization only once, allowing for sample generation from different causal models without re-training.
> # Weakness 4
> Following the reviewer’s recommendations, we used the CNF package to estimate ATEs in our synthetic data experiments with different causal effects, comparing the results against frugal flows and a causal statistical matching algorithm. These results are presented in Table 1 in the attached document. CNFs showed higher variance in ATE estimates than frugal flows. In addition, we also provided figures showing the observational likelihood loss during model training for both real-world datasets in Figures 4 & 5.
> # Additional Questions:
> * The dotted red line represents the customized ATE parameter of the frugal flow, indicating that causal inference algorithms infer the specified ATE. We will clarify this in the updated version of the paper.
> * Lines 291-294 indicate that frugal flows are not recommended for inferring ATEs in small to medium-sized datasets. Figure 3 in the attached experimental results showing that frugal flows struggle to converge to the true ATE of synthetic data when $N < 10,000$. However, for benchmarking, frugal flows can customize the causal properties of generated data and generate realistic samples with specified ATEs even for smaller datasets like Lalonde.
> * We agree that Figure 1 should be placed closer to section 2.2 and will make this change in the final version.

---

> > ### Comment · Reviewer_PATR · 2024-08-11
> >
> > I thank the authors for their super detailed responses and the additional experiments. I do think they add quite some value, and if the authors use the feedback to improve the readability of the manuscript (and thus how welcoming it is for newer audiences), I think it has potential to be quite a good paper.
> >
> > With respect to the new results, I am a bit surprised by the results of CNF on model 1. Just to make sure, I'd encourage the authors to check that they used a model different from MAF for the base model, as this is not a universal density approximator.
> >
> > In any case, I am happy with the response from the authors and I will update my score.

---

### Author Rebuttal · Authors · 2024-08-07

We thank all the reviewers for their comprehensive commentary and very helpful suggestions for our paper. In addition to the individual responses to each reviewer, we provide a more global summary of what we believe were the core themes across all four reviews. These centre around 1) clarity and terminology, 2) further inference experiments (following Section 4.1 in the original submission) and 3) a more comprehensive validation of the synthetic data generation in Section 4.2, showing that frugal flows indeed generate realistic data samples which resemble the original dataset.
# Clarity and Terminology
Some reviewers suggested to increase clarity and precision in our definitions and terminology. In response, we have made the following changes:
1) **Definitions and Assumptions**: We are happy to add more detailed explanations and explicitly define terms such as conditional ignorability and variation independence. In addition, we wish to clarify the assumptions required for frugal flows; in particular, the covariate set must only include pretreatment covariates
2) **Clarification on Frugal Parameterization**: We will elaborate on our explanation of the frugal parameterization and more acutely describe how targets the causal margin rather than the conditional effect. This includes an expanded discussion on the relevance of copulas and how they enable frugal parameterization to capture the causal margin effectively.
# Further Inference Experiments
Reviewers expressed a desire for more extensive inference experiments, expanding on the contents in Section 4.1 in the original submission. We have addressed this by:
1) **Benchmarking Against Comparable Methods:** We are grateful for the additional references provided by the reviewers, in particular the Causal Normalizing Flows (CNFs) paper, which we are happy to add to the paper. We reran the experiments in Section 4.1 and use CNFs to estimate the ATEs in simulated datasets. We use the hyperparameter settings recommended in the package and do not perform hyperparameter tuning. For the flow architecture, we use Neural Spline Flows, as the paper reports in Appendix D.3 that they yield a better performance than simple Masked Autoregressive Flows. In addition to a (misspecified) linear outcome regression in the original submission, we use an implementation of a causal statistical matching algorithm to add further variety to the inference algorithms
2) **Expanded Experiments**: We conducted experiments using simulated data generated from two true ATEs (1 and 5) using models $M_{1}$ and $M_{2}$ described in Section 4.1 in the original submission ($M_{3}$ omitted for now due to time constraints, but we will update the paper with the results of all three models). The results, which are presented in Table 1 in the attached material, show that frugal flows consistently identified the true ATE with the lowest standard error. CNFs performed worse for data generated in $M_{1}$, demonstrating a clear bias, but correctly estimated the true ATE in data generated from $M_{2}$, with a lower standard deviation than statistical matching but higher than frugal flows.
# Validation of Synthetic Data Generation
We agree with the reviewers that comprehensive validation of our approach is crucial. To this end, we make the following comments:
1) **Contrasting against Credence and RealCause**: We have enhanced the validation section by adding a discussion contrasting Credence (Parikh et al, 2023) and RealCause (Neal et al, 2021), the two most similar existing methods for generating realistic synthetic data for validating causal inference algorithms. Credence allows the user to exactly specify what the conditional ATE is in the underlying data, much in the same way as frugal flows allow the user to exactly specify the ATE. RealCause only allows the user to scale the causal effect posthoc, which in particular hinders its ability to model a null hypothesis (i.e all observations will be set to zero); this is straightforward for frugal models (see Section 3 in Evans and Didelez, 2024). Furthermore, frugal flows’ ability to model unobserved confounding is shared with Credence and not with RealCause. For these reasons we chose Credence as a more appropriate method to compare experimentally against frugal flows, and discuss these experiments in the next paragraph.
2) **How Realistic are the Synthetic Data?:** We compared frugal flows against Credence using both the Lalonde and e401(k) datasets from our original submission. We present correlation matrices of pretreatment covariates and outcomes for the original data against samples generated from frugal flows and Credence trained with different causal constraints.These results are shown in Figures 1 and 2 in the attached document. Frugal flows generated samples closely resembling the original data, particularly for the larger e401(k) dataset. While Credence performed well in generating realistic samples, altering the causal effect rigidity impacted the covariate joint and introduced dependencies not present in the original data. A key advantage of frugal flows is the need to optimize the model fitting only once, allowing for direct modification of causal constraints without affecting the covariate joint and propensity score.
3) **Multivariate Statistical Tests:** We conducted a variety of multivariate statistical tests (Maximum Mean Discrepancy test, energy test, Friedman-Rafsky test, and k-nearest neighbor test) to assess whether the synthetic data from the generative models is statistically similar to the original training data. These results are presented in Table 2 of the attached document, and suggest that frugal flows do indeed generate realistic looking datasets.

Once again, we thank the reviewers and the area chair for their consideration of our paper, and we are happy to answer any follow up questions you may have.

---

### Decision · Program_Chairs · 2024-09-25

**Decision:**

Accept (poster)

**Comment:**

Summary:

The paper introduces Frugal Flows, a generative modeling approach that learns the data generation process with explicit parametrization for the marginal causal effect of treatment on outcomes.

Pros:

+ Addresses a significant gap by allowing for general constraints on quantities of interest during data generation for causal inference.

+ Novel approach combining frugal parametrization, normalizing flows, and copulas for causal effect estimation and synthetic data generation.

+ Well-written with clear explanations of complex concepts and potential for high impact in the causality research community.

Cons:

+ Limited empirical validation compared to prior work in generative modeling for causal inference.